# Transfer Entropy Bottleneck:
# Learning Sequence to Sequence Information Transfer

**Damjan Kalajdzievski**[1,2,*]                    *damjank7354@gmail.com*

**Ximeng Mao**[1,3,*]                    *ximeng.mao@mila.quebec*

**Pascal Fortier-Poisson**[4]                    *pascal@bios.health*

**Guillaume Lajoie**[1,5,8,†]                    *g.lajoie@umontreal.ca*

**Blake A. Richards**[1,2,6,7,8,†]                    *blake.richards@mila.quebec*

[1] *Mila - Quebec AI institute, Montreal, QC, Canada*
[2] *Department of Neurology and Neurosurgery, McGill University, Montréal, QC, Canada*
[3] *Department of Computer Science and Operations Research, Université de Montréal, Montréal, QC, Canada*
[4] *BIOS Health Ltd., Cambridge, UK*
[5] *Department of Mathmatics and Statistics, Université de Montréal, Montréal, QC, Canada*
[6] *School of Computer Science, McGill University, Montréal, QC, Canada*
[7] *Montreal Neurological Institute, McGill University, Montréal, QC, Canada*
[8] *CIFAR, Toronto, ON, Canada*
[*] *Equal Contribution*
[†] *Equal Advising*

**Reviewed on OpenReview:** *https://openreview.net/forum?id=kJcwlP7BRs*

## Abstract

When presented with a data stream of two statistically dependent variables, predicting the future of one of the variables (the target stream) can benefit from information about both its history and the history of the other variable (the source stream). For example, fluctuations in temperature at a weather station can be predicted using both temperatures and barometric readings. However, a challenge when modelling such data is that it is easy for a neural network to rely on the greatest joint correlations within the target stream, which may ignore a crucial but small information transfer from the source to the target stream. As well, there are often situations where the target stream may have previously been modelled independently and it would be useful to use that model to inform a new joint model. Here, we develop an information bottleneck approach for conditional learning on two dependent streams of data. Our method, which we call Transfer Entropy Bottleneck (TEB), allows one to learn a model that bottlenecks the directed information transferred from the source variable to the target variable, while quantifying this information transfer within the model. As such, TEB provides a useful new information bottleneck approach for modelling two statistically dependent streams of data in order to make predictions about one of them.

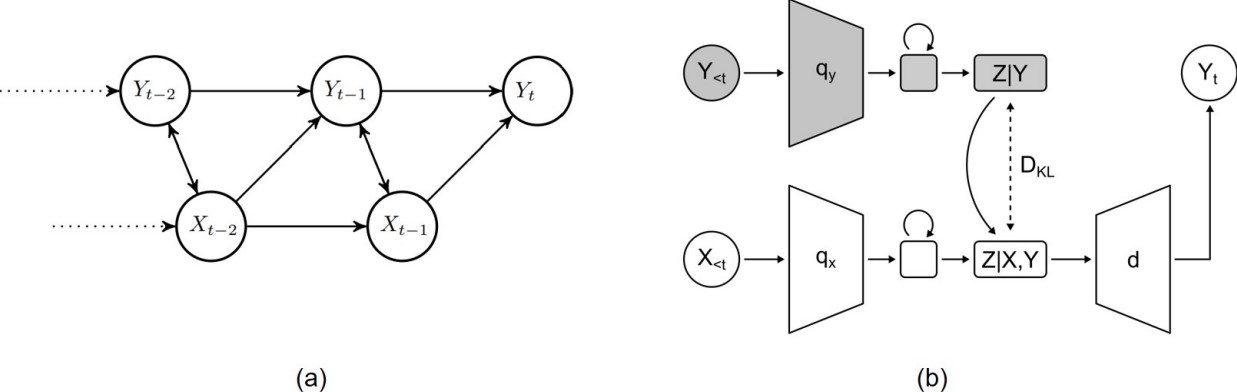

Figure 1: **(a)**: Probabilistic graph representing directed information transfer between stochastic processes. $X$ is the source stream, $Y$ is the target stream. **(b)**: Architecture diagram of the TEB model as implemented.

# 1 Introduction

Scientists are often presented with two streams of statistically coupled data, a source stream, which provides information, and a target stream, which they wish to predict the future of (See Figure 1a). Obviously, one can use the history of an individual variable to predict its future values, but at the same time, it is desirable to use the statistical dependency in a joint model, rather than treating each variable as independent. However, using a typical joint input stream model ignores the directed nature of the information transfer from the source stream to the target stream. Put another way, one would ideally use a method that can differentiate the directionality of information in order to better modulate and interpret the role of different input streams on prediction. To this end, we propose to leverage **transfer entropy** (Schreiber, 2000), a measure of directed information flow between stochastic processes, to help with the modelling. Specifically, if we have a source stream, $X_{t-\ell}, \ldots, X_t$, and a target stream, $Y_{t-\ell}, \ldots, Y_t$, the transfer entropy from $X$ to $Y$ (with horizon $0 < \ell \leq \infty$) is:

$$T_{X \xrightarrow{t} Y} = I(Y_t; (X_i)_{t-\ell \leq i < t} | (Y_i)_{t-\ell \leq i < t})$$

where $I(\cdot; \cdot | \cdot)$ is the conditional mutual information.

Estimating transfer entropy with a deep learning approach has been extensively explored (Zhang et al., 2019) (see Appendix C for discussion contrasting with TEB). But, leveraging the transfer entropy for prediction is often a challenging problem and one that has yet to be resolved for deep neural networks (Schreiber, 2000; Gençağa, 2018). One reason for this difficulty is that the transfer entropy can be small relative to the overall entropy in the data streams, and as such, many prediction models will ignore it. For example, in weather forecasting, periodicity of average monthly temperatures could dominate a prediction model of future temperatures, which would prevent a simple predictive model from detecting important anomalies indicated in other sources of information such as air pressure, wind, greenhouse gas emission and other human activities. Indeed, we shall show that a neural network trained on a joint input stream can struggle to learn or generalize when the statistical coupling between the two processes is strongly correlated, but with a conditionally small and crucial information transfer from the source stream.

We use an information bottleneck (IB) approach (Tishby et al., 2000) to derive a method, which we term **Transfer Entropy Bottleneck (TEB)**, that learns a compressed conditional information representation. This allows the benefits of IB to be applied to the new domain of double stream conditional processing. As in other information bottleneck architectures, TEB works by learning a latent representation of the data, $Z$, such that $Z$ contains only the relevant information required to capture the relationship between the random processes $X$ and $Y$ (Tishby et al., 2000). But, in TEB, this is done using information that is *conditional on*

*the history of* $Y$, allowing one to capture both the statistical dependencies over time on $Y$, as well as the transfer entropy. As well, we design TEB in a manner that allows one to use a pre-trained predictive model for the target stream. That is, if one has an existing model that has been trained to predict $Y_t$ from the history of $Y$ (i.e. $(Y_i)_{t-\ell \le i < t}$), TEB allows one to plug this existing model into a new model that includes $X$ in order to capture the transfer entropy and use it to improve the prediction of $Y_t$. This provides a flexible solution when one might have reasons to keep a previously trained model fixed. In practice, the reasons can be the desirable performance and interpretability of a given model, retraining being computationally expensive and time-consuming, and/or compatibility to other components in a larger system.

More precisely, the contributions of this paper are the following:

- We design a loss function that promotes representations obeying a **Conditional Minimum Necessary Information (CMNI)** principle, which ensures that the latent representation of the data contains only the relevant information transferred from the source to the target stream. We use this loss to develop TEB, an information bottleneck algorithm that learns representations following the CMNI, such that:
  1. The learnt representation captures information embedded in the transfer entropy.
  2. The target stream can have its latent representation pre-trained separately from the joint latent representation, which can provide numerous computational and other practical benefits.

- We introduce three synthetic tasks on dual stream modeling problems. We provide experiments on these tasks to show that TEB allows one to improve the predictions of the target stream, and that it is applicable to various data modalities including images and time-series signals.

Altogether, TEB is a mathematically principled new IB method for learning to predict from two streams of statistically dependent data with a wide array of potential applications.

## 1.1 Related work

The concept of an IB, as first defined in Tishby et al. (2000), was introduced as a way of quantifying the "quality" of a compressed representation, $Z$, of a signal, $X$, for making inferences about some other signal, $Y$. This work defined high quality latent representations as being those that minimize the following Lagrangian:

$$\text{IB} = I(X; Z) - \beta I(Y; Z). \tag{IB}$$

In Tishby & Zaslavsky (2015), the IB concept is explored in deep neural networks as a possible approach for studying the quality of representations in the hidden layers of a network, and as a possible explanation for generalization. The analysis of an IB so as to understand the learned representations of neural networks was further studied in Saxe et al. (2019).

Traditionally, the IB literature seeks to find a model with the most compressed representation, $Z$, that preserves the relevant information about $Y$. This approach seeks out a representation where the following equality holds:

$$I(X; Z) = I(Y; Z). \tag{MNI}$$

We use the terminology of Fischer (2020), and say that such a representation captures the **Minimum Necessary Information (MNI)**. In the case of IB variable $Z$ minimizing the information about $X$ relevant to $Y$, this is the same as $Z$ being a minimal sufficient statistic of $X$ for $Y$. Note that it might not be possible to exactly reach MNI (Wu et al., 2019; Fischer, 2020), however, it proves to be a valid training objective in practice (Fischer, 2020).

The previously developed IB methods most relevant to ours, are the **Variational Information Bottleneck (VIB)** (Alemi et al., 2017) and the **Conditional Entropy Bottleneck (CEB)** (Fischer, 2020).

Deep VIB (Alemi et al., 2017) is a method using a latent variable encoder-decoder model to derive a variational IB procedure, with a training approach analogous to variational autoencoders (Kingma & Welling, 2014). They show this procedure can be used to learn an IB representation in neural networks. Put more precisely, with the computation arranged in a feed-forward structure $X \to Z \to Y$, via an encoder $q(z|x)$ and a decoder $d(y|z)$, VIB optimizes $q, d$ with a variational bound for the IB loss:

$$\text{VIB} = \mathbb{E}_{p(x)}\left[D_{\text{KL}}(q(z|x)\|p(z))\right] - \beta \mathbb{E}_{p(y,z)}[\log(d(y|z))] \tag{VIB}$$

where $p(z)$ is a pre-specified fixed prior. In this loss, the KL divergence term bounds the information that $Z$ carries about $X$:

$$I(Z;X) \leq \mathbb{E}_{p(x)}\left[D_{\text{KL}}(q(z|x)\|p(z))\right],$$

and so minimizing this term compresses the representation, while the other term:

$$\mathbb{E}_{p(y,z)}[\log(d(y|z))] \leq I(Y;Z)$$

is maximized to increase the information that $Z$ carries about $Y$. Alemi et al. (2017) show that their method has increased generalization and adversarial robustness compared to other forms of regularization like dropout (Srivastava et al., 2014), and confidence penalty with label smoothing (Pereyra et al., 2017).

The CEB method (Fischer, 2020), proceeds similarly to VIB, but instead of using the KL divergence to a prior as a bound for $I(Z;X)$, it minimizes an upper bound of the difference $I(Z;X) - I(Y;Z)$ using a backwards encoder $b$. The loss minimized for $q, d, b$ is then:

$$\text{CEB} = \mathbb{E}_{p(y,x)}\left[D_{\text{KL}}\left(q(z|x)\|b(z|y)\right)\right] - \gamma \mathbb{E}_{p(y,z)}[\log(d(y|z))]. \tag{CEB}$$

TEB shares many conceptual links with both VIB and CEB. However, unlike VIB and CEB, TEB is specifically designed to work with situations where one has *two* sequences of data, one serving as a source stream and one as a target stream. Thus, TEB can be thought of as a means to bring similar principles of the IB and MNI in order to obtain a high quality representation that captures directed information transfer.

In Skatchkovsky et al. (2021), a variational information bottleneck method is derived for directed information (Kramer, 1998), which is a measure similar to transfer entropy. In contrast to ours, their method uses a non-conditional prior, and imposes stronger Markov assumptions. In implementation, the prior is a fixed distribution, they limit their approach to simple spiking network encoders, and only process non-sequential data. Thus, though related, TEB solves a very different problem. Note that throughout the paper we use the term "directed information transfer" to denote transfer entropy, not to be confused with "directed information" above which is a separate measure.

Another related concept to transfer entropy is Granger causality (Granger, 1969), and many previous works have focused on the relationship between the two (Barnett et al., 2009; Hlavackova-Schindler, 2011). For example, Barnett et al. (2009) showed that the two concepts are equivalent under Gaussian assumptions. Originally developed in econometrics, Granger causality has gained attention in other domains, including neuroscience (Seth et al., 2015) and industrial processes (Lindner et al., 2019). The calculation of Granger causality is traditionally via linear vector autoregressive models (VAR), which is generally considered cheaper than transfer entropy. To estimate Granger causality of X to Y in practice, VAR trains two linear regression models to predict Y with or without X as input, and Granger causality can be calculated as a measure on how much the latter reduces the error of the former. Throughout the years, many works extended this traditional approach in terms of both causality analysis from the data and time-series forecasting, for the linear model (Lozano et al., 2009), as well as non-linear models including kernel regression (Gregorová et al., 2017), radial basis functions neural network (Wismüller et al., 2021), multilayer perceptrons (Talebi et al., 2017; Tank et al., 2022) and recurrent neural networks (Tank et al., 2022). In general, these methods aim to learn the causal relationships among different time-series and / or among different time-lagged variables in a regression problem, and that is very different from the objective of TEB, and other IB methods described above, which is to learn a compressed latent representation of the data.

## 2 Methods

In this section we fully specify TEB for learning representations which compress the transfer entropy via a bottleneck. Note that conditional mutual information is a special case of transfer entropy, and so TEB is also an IB method for creating a conditional information bottleneck. We encourage the interested reader to read the Appendix A for a thorough exposition of the mathematical details and proofs.

Our situation is analogous to a latent variable encoder-decoder model, but with a conditional structure; We wish to learn a latent encoding $Z_t$ of $(X_i)_{t-\ell \leq i < t}$ via a map $q$, which is conditional on $(Y_i)_{t-\ell \leq i < t}$, from which one is able to decode $Y_t$ conditionally on $(Y_i)_{t-\ell \leq i < t}$ with the decoder $d$. Put another way, we want to be able to predict $Y$ at time $t$ using the values of $Y$ and $X$ up to time $t-1$, but using our latent encoding of $X$ conditioned on the history of $Y$. Namely, denoting $Z = Z_t$, $X = (X_i)_{t-\ell \leq i < t}$, $Y = (Y_i)_{t-\ell \leq i < t}$, and $Y' = Y_t$,, we have $Z, p, q$ such that:

$$p(y', z, x, y) = p(y', x, y)p(z|x, y) = p(y', x, y)q(z|x, y), \tag{1}$$

as illustrated in Figure 2a. Note that $q(z|x, y)$ is actually the true distribution $p(z|x, y)$, since we artificially introduced the variable $Z$ as defined by $q$. Note that Equation 1 is equivalent to the assumption on the conditional independence $Y' \perp Z|X, Y$.

Furthermore, we require the distribution to be learned by a feed-forward encoder-decoder structure, which assumes the following decomposition of joint probability distribution (as illustrated in Figure 2b), where $d(y'|z, y)$ is a variational approximation to $p(y'|z, y)$:

$$p(y', z, x, y) = p(x, y)p(y'|z, y)p(z|x, y) \approx p(x, y)d(y'|z, y)q(z|x, y). \tag{2}$$

The first part of Equation 2 is equivalent to the conditional independence $Y' \perp X|Z, Y$.

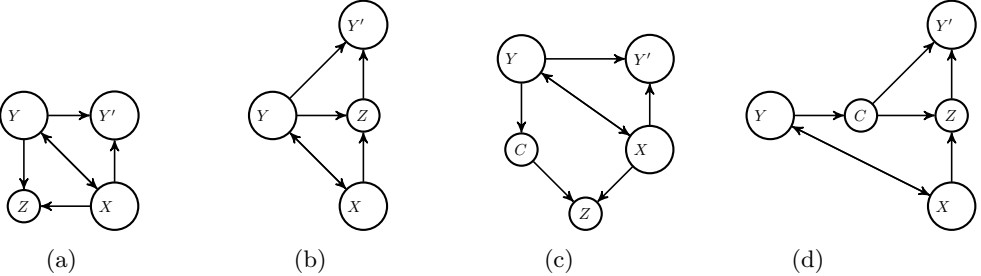

(a)  (b)  (c)  (d)

Figure 2: Probabilistic graph representing the random variables and their relationships as assumed by TEB. **(a)**: The actual random variables in the dataset. Independencies include $Y' \perp Z|X, Y$. **(b)**: The random variables as computed by our feed-forward computation. Independencies include $Y' \perp X|Z, Y$. **(c)**, **(d)**: The actual and feed-forward graphs, respectively, when utilising a pre-trained context encoding $C$. Independencies for (c) include $Y' \perp Z|X, C$, $Y' \perp C|Y$, and $Y' \perp C|X, Y$, and for (d) include $Y' \perp X|Z, C$ and $Y' \perp Y|X, C$.

As is standard in IB methods, we want our optimization procedure to target a MNI principle of optimal compression; i.e. we wish to learn our latent representation, $Z$, such that it captures exactly the necessary conditional information (conditioned on $Y$) between $X$ and $Y'$. Specifically we would like to target the following analogous MNI point, the CMNI point:

$$I(Y'; X|Y) = I(Y'; Z|Y) = I(Z; X|Y). \tag{CMNI}$$

Due to the more complex probabilistic structure of our setting, we do not have $I(Y'; Z|Y) \leq I(Y'; X|Y)$ and $I(Y'; X|Y) \leq I(Z; X|Y)$ directly from the data processing inequalities, as opposed to the situation in

the standard IB setting, where the joint probability follows a simpler Markov chain $Z \to X \leftrightarrow Y$. We thus prove that this is the case given our set-up (see Proposition 3 in Appendix A), which implies that we can arrive at the CMNI point by maximizing $I(Y'; Z|Y)$ and minimizing $I(Z; X|Y)$. We remark here that, as in reaching the MNI point for IB methods, it is potentially not possible in practice to reach the CMNI point exactly on a given dataset. However, precisely reaching the CMNI point does not appear to be a necessity for effective compression in practice as long as one approaches it.

However, estimating directly $I(Y'; Z|Y)$ and $I(Z; X|Y)$ can be challenging, so we derive variational bounds to optimize the two conditional mutual information terms.

First, to maximize $I(Y'; Z|Y)$ we decompose this information as

$$I(Y'; Z|Y) = -H(Y'|Z, Y) + H(Y'|Y) \propto -H(Y'|Z, Y). \tag{3}$$

Importantly, the term $H(Y'|Y)$ is fully determined by the true data distribution, and thus, can be dropped from the optimization procedure, which is why the proportionality holds.[1]

We can readily bound $-H(Y'|Z, Y)$ from below with:

$$\begin{aligned}
-H(Y'|Z, Y) &= \mathbb{E}_{p(y,z,y')}[\log(p(y'|z, y))] \\
&= \mathbb{E}_{p(z,y)}\left[D_{\mathrm{KL}}\left(p(y'|z, y)\|d(y'|z, y)\right)\right] + \mathbb{E}_{p(y,z,y')}[\log(d(y'|z, y))] \\
&\geq \mathbb{E}_{p(y,z,y')}[\log(d(y'|z, y))].
\end{aligned} \tag{4}$$

Thus, in order to maximize $I(Y'; Z|Y)$ we can simply maximize $\mathbb{E}_{p(y,z,y')}[\log(d(y'|z, y))]$. Moreover, since maximizing this bound also minimizes the expected KL divergence, i.e.

$$\mathbb{E}_{p(y,z,y')}[\log(p(y'|z, y))] - \mathbb{E}_{p(y,z,y')}[\log(d(y'|z, y))]$$

, we will simultaneously also learn to approximate $p(y'|z, y)$ with $d(y'|z, y)$, and the bound will become tight.

To minimize $I(Z; X|Y)$ by a tractable upper bound one needs to do more work, and use more distributional approximations. In VIB, one bounds $I(Z; X)$ above with $I(Z; X) \leq \mathbb{E}_{p(x)}\left[D_{\mathrm{KL}}(q(z|x)\|p(z))\right]$, where $p(z)$ is a pre-specified fixed prior. We shall see that $I(Z; X|Y)$ can be bounded above similarly for TEB, however our prior cannot be a hand-set fixed distribution, since it must be relevant to $Y$. Introducing the distribution $q_y(z|y)$ as a learnable approximation to $p(z|y)$, and recalling that $q(z|x, y) = p(z|x, y)$ is the true distribution, we get that:

$$\begin{aligned}
I(Z; X|Y) &= \mathbb{E}_{p(y,z,x)}\left[\log\left(\frac{q(z|x, y)}{p(z|y)}\right)\right] \\
&= \mathbb{E}_{p(y,z,x)}\left[\log\left(\frac{q(z|x, y)}{q_y(z|y)}\right)\right] - \mathbb{E}_{p(y)}\left[D_{\mathrm{KL}}(p(z|y)\|q_y(z|y))\right] \\
&\leq \mathbb{E}_{p(y,z,x)}\left[\log\left(\frac{q(z|x, y)}{q_y(z|y)}\right)\right] = \mathbb{E}_{p(x,y)}\left[D_{\mathrm{KL}}(q(z|x, y)\|q_y(z|y))\right].
\end{aligned} \tag{5}$$

Training $q_y(z|y)$ by minimizing this bound, also gives $q_y(z|y) \approx p(z|y)$, since it squeezes the KL divergence of the two, and so separate learning of $q_y$ to approximate $p(z|y)$ is not necessary.

Equation 4 with Equation 5 is used to define a Lagrangian loss for the TEB algorithm, which can be shown to bring us to the CMNI point when minimized:

**Theorem 1.** *Under the independence assumptions implied by Equation 1 and 2, the minimum of the following objective with respect to $q, q_y, d$ will be at the CMNI point for $Z, X, Y, Y'$, for some hyperparameter $\beta > 0$:*

$$TEB = \mathbb{E}_{p(x,y)}\left[D_{\mathrm{KL}}(q(z|x, y)\|q_y(z|y))\right] - \beta\mathbb{E}_{p(y,z,y')}[\log(d(y'|z, y))]. \tag{TEB}$$

---

[1]However, this also means that the information $I(Y'; Z|Y)$ may not be possible to explicitly calculate from our existing procedure.

*Proof.* Directly from Proposition 3 in Appendix A.1, the CMNI point can be arrived at by maximizing $I(Y'; Z|Y)$ and minimizing $I(Z; X|Y)$. The proof of Theorem 1 is immediate from the proposition, by replacing $I(Y'; Z|Y)$ with its lower bound in Equation 4 and $I(Z; X|Y)$ with its upper bound in Equation 5. Please refer to Appendix A.1 and A.2 for the full derivations of Theorem 1.

Moreover, at convergence, the value of the upper bound is given by:

$$\mathbb{E}_{p(x,y)}\left[D_{\mathrm{KL}}(q(z|x,y)\|q_y(z|y))\right] \approx I(Z; X|Y) \tag{6}$$

and so it can be used as a metric for $I(Z; X|Y)$. Thus in the case that the TEB loss optimum is at the CMNI point, we also have the fact that this bound measures the transfer entropy $I(Y'; X|Y)$. Note that if the decoder does not converge to the true decoder, the bounds we use for optimization are still valid, and all we stand to lose is the ability to get to the CMNI point exactly. No matter how bad the decoder is, for whatever output space we are able to represent, we still do bottleneck $I(Z; X|Y)$, and our bound of $I(Z; X|Y)$ becomes tight if converged.

The loss function of TEB shares structural similarity with those of VIB and CEB but differs in the choice of prior distribution in the Kullback-Leibler (KL) divergence term. The VIB method uses a fixed distribution as the prior (a Gaussian in the original implementation), while for CEB, the prior is learned from the image of a target $Y'$, mapped back to the latent via a backwards encoder. In contrast, TEB learns the prior only from the history of $Y$, making the role of the KL term to regularize the jointly inferred latent $Z$, so that it relies on as little information from the $X$ stream as possible.

Now, suppose one has already learned an encoding $q_y(\cdot|y)$ for $Y$ that provides a latent representation of $Y$. We can use this as a fixed prior in TEB. This could be any existing model for encoding $Y$ to a variable $C$, e.g. a pre-trained self-supervised model, which will not be optimised with the derived TEB optimisation procedure (see Figure 2c and 2d to see how we integrate this latent in the TEB computation). One can then use this encoding in place of $Y$ in the TEB algorithm. This could come with many practical advantages like computational efficiency, training efficiency, reuse of existing models, and rapid prototyping when experimenting with multiple configurations or hyperparameter settings for the full TEB model.

To successfully bottleneck the transfer entropy with respect to conditioning on $Y$ instead of $C$, it suffices to require that the representation $C$ is relevant to $Y'$ in that $I(C; Y')$ is maximized (see Lemma 5 in Appendix A), so that $I(X; Y'|C) \approx I(X; Y'|Y)$. With this in mind we get the following TEB loss for CMNI learning from a pre-existing contextual encoding:

**Theorem 2.** *Under the independence assumptions $Y' \perp Z|X, C$, $Y' \perp C|Y$, $Y' \perp C|X, Y$, $Y' \perp X|Z, C$ $Y' \perp Y|X, C$, and $C \sim q_y(\cdot|y)$ such that $I(C, Y') = I(Y, Y')$, the minimum of the following objective with respect to $q, d$ will be at the CMNI point for $Z, X, Y, Y'$, for some hyperparameter $\beta > 0$:*

$$TEB^c = \mathbb{E}_{p(x,y)}\left[D_{\mathrm{KL}}(q(z|x,c)\|q_y(z|y))\right] - \beta\mathbb{E}_{p(y,z,y')}[\log(d(y'|z,c))]. \tag{TEB$^c$}$$

*Proof.* Directly from Lemma 5 in Appendix A.3, with $I(C; Y') = I(Y; Y')$, we can replace $Y$ everywhere with $C$ and still arrive at the CMNI point. Then the proof is immediate by following a similar procedure as Theorem 1. Please refer to Appendix A.3 for the full derivations of Theorem 2.

## 3 Experiments

To implement TEB we used an architecture as shown in Figure 1b. It should be noted that TEB could be applied to other architectures, but here we will describe the architecture used in our experiments. The model uses two initial encoders for the two streams $q_y$ and $q_x$, both of which feed into two separate LSTMs (Hochreiter & Schmidhuber, 1997) to aggregate information across the sequence. The hidden state from the $Y$ pathway LSTM is then linearly projected to calculate the means and log variances for the dimensions of the prior $q_y(z|y)$, which are treated as being independent Gaussians in each dimension. The mean and log variance of this prior are then combined with the output from the $X$ pathway LSTM, via a multi-layer perceptron (MLP) that perturbs the prior $q_y$, to arrive at means and log variances parameterizing

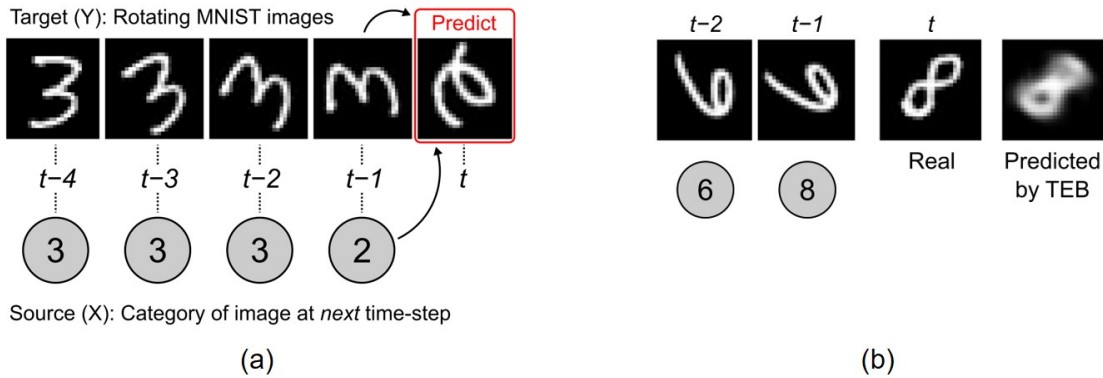

Figure 3: **(a)**: Sample sequence of rotating MNIST, which switches example digit at the fourth frame. **(b)**: This is a sample testing generation of a TEB model using a fixed pre-trained context encoding, reconstructing the next step of a sequence which always switches digits at the prediction step. Images are arranged from left to right as: two frames of input video, next step, generated prediction. As an added demonstration of flexibility of TEB in this figure, the pre-trained context encoding was trained to reconstruct the next step of a rotating digit. The dataset the pre-trained context encoding was trained on is a separate one without switching of digits, hence the context encoding model had never encountered on video where the digit example switched. Note that the blur in the prediction is expected since the optimal prediction is the least squares minimum distance to all examples of the corresponding class.

the independent Gaussians in each dimension of $q(z|x, y)$. Then the reparameterization trick (Kingma & Welling, 2014) is used to sample latent representation $Z$ from $q(z|x, y)$.

When the expected decoder $d$ output is an image, the decoder arranges the sampled latent representation $Z$ into a spatial representation using a positional encoding, and this version of the latent is then transformed via space preserving convolutions to arrive at the output image, $Y'$. When the decoder is used to generate a time-series signal, it is implemented as a neuralODE (Chen et al., 2018) with its dynamics parameterized by an MLP. In both cases, we assume the decoder output is the mean of a Gaussian distribution with fixed variance, independently in each pixel or at each time-step. See Appendix B for exact specifications of model architecture, more detailed descriptions on the task datasets, as well as more sample generations in every experiment. Code of our implementations can be found at `https://github.com/ximmao/TransferEntropyBottleneck`.

## 3.1 Generating Rotated MNIST

To experimentally test Theorems 1 & 2, we create a dataset of videos of rotating MNIST (Lecun et al., 1998) digits. In a given video sequence, a digit starts rotated at some random angle that is a multiple of $\frac{\pi}{4}$, and rotates counter-clockwise $\frac{\pi}{4}$ at every frame, but with a chance that the example digit in the video may change to another random digit at a given frame with some specified probability. We can interpret this data as a target stream $Y$ which is the image video, and source stream $X$ consisting of class labels for the next step in the video. See Figure 3a for an illustration of the rotating MNIST task.

To test the properties of TEB near the CMNI point, we train several models with different values of $\beta$ for 10 epochs. On this test we set a next step digit switching probability of 0.5, which gives that the true expected transfer entropy from $X$ to $Y$ is 1.67689 nats. See Appendix B.2 for the calculation of the true transfer entropy for this task.

From Figure 4a, we can see that there is a "bifurcation point" for $\beta$ around 0.2, such that for any $\beta$ smaller than that the output loglikelihood sharply drops off. We refer to such a point as $\beta_1$, and the sharp drop-off for $\beta < \beta_1$ is likely due to that essential information transfer from X being left out of the latent representation Z of the model, as in $I(Z; X|Y) < I(Y'; X|Y)$. On the other hand, as in other IB methods (Wu et al., 2019), there is another value of $\beta$, which we denote $\beta_0$ as in Wu et al. (2019), such that for $\beta < \beta_0$ the model is not learning and $Z$ becomes a trivial representation, which in our case means that we compressed too much on

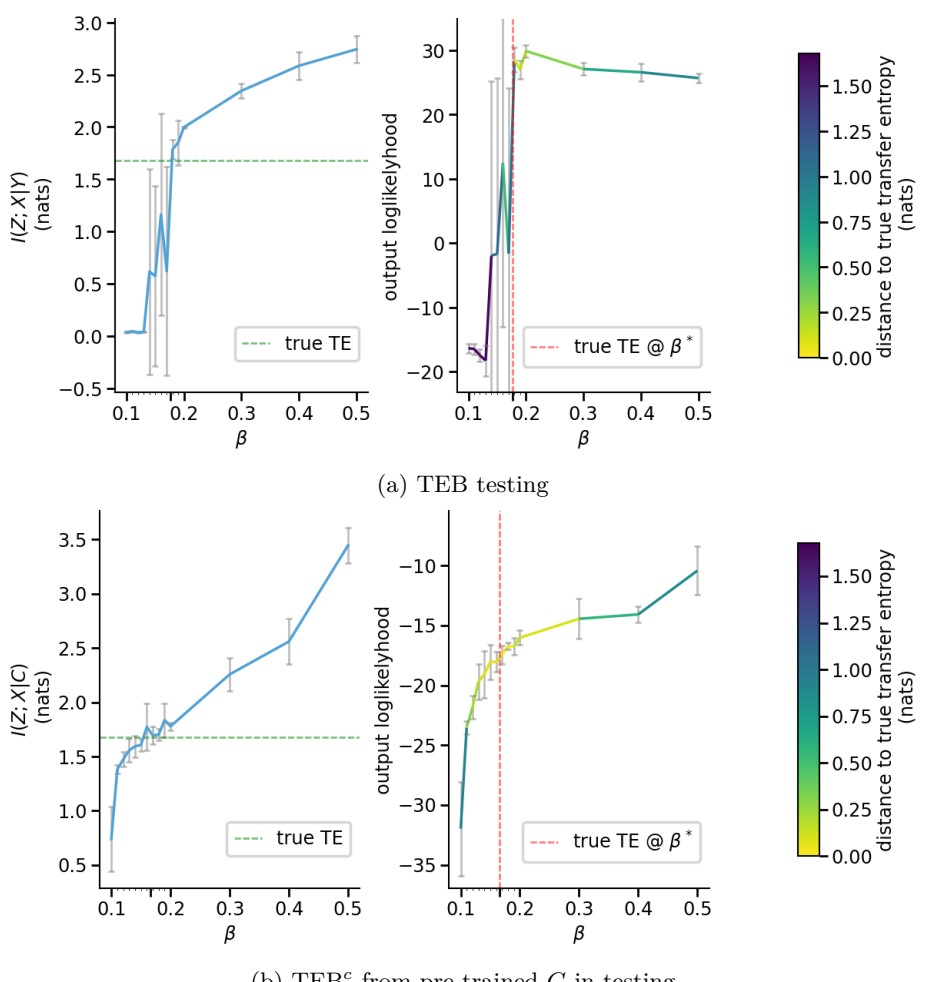

(a) TEB testing

(b) TEB$^c$ from pre-trained $C$ in testing

Figure 4: Plots of the information metric and reconstruction loglikelihood vs $\beta$ incremented by tenths, with extra hundredths increments in the chaotic regime in $[.1, .2]$, for both TEB and TEB$^c$ testing. In the information metric subfigure (the left subfigure in both (a) and (b)), the green dashed horizontal line indicates the true transfer entropy for this dataset. From this we obtain $\beta^*$ as the specific value of $\beta$ achieving the true transfer entropy, calculated as the interpolated value of $\beta$ for which $I(Z; X|Y) = I(Y'; X|Y)$ on average from repeated trials. In the reconstruction loglikelihood subfigure (the right subfigure), the output loglikelihood of the model trained with the obtained $\beta^*$ is marked using the red dashed vertical line. Note that in (a) the variance of runs highly increases for $\beta \in (.1, .2)$, but the models in this interval that found a solution with $I(Z; X|Y) \approx I(Y'; X|Y)$ are the same models that achieved higher reconstruction performance.

the information in the bottleneck, so that the information from X to recover Y' is not available to the model any more. From Figure 4a, we can put $\beta_0$ roughly in the range from 0.1 to 0.14, corresponding to nearly zero $I(Z; X|Y)$. Note that $\beta_0, \beta_1$ are "bifurcation points" for the task performance both in training and testing (see Appendix B.2 for training plots), as for values of $\beta \in (\beta_0, \beta_1)$ we quickly begin to learn representations that perform better until $\beta \geq \beta_1$ and the model does not compress lower than the true transfer entropy $I(Y'; X|Y) \leq I(Z; X|Y)$. In Figure 5, we show the image generation samples when varying $\beta$.

We choose to mark $\beta^*$ in Figure 4 as the interpolated value of $\beta$ for which $I(Z; X|Y) = I(Y'; X|Y)$ on average. At $\beta^*$ the true expected transfer entropy is approximately identifiable by the value of our upper bound for $I(Z; X|Y)$ in Equation 6, and we find that $\beta^*$ lies close to $\beta_1$. We can see that the models learned

at $\beta_1$ are very close to the CMNI point with $I(Z; X|Y) \approx I(Y'; X|Y) \approx I(Y'; Z|Y)$. [2] Consistent with IB principles, we can see that at values of $\beta$ which result in the representation closest to the CMNI point, TEB obtains approximately the best testing score. Moreover, on this dataset the TEB model testing performance decreases monotonically with distance of $I(Z; X|Y)$ from $I(Y'; X|Y)$, outside of a small neighbourhood of $\beta_1$.

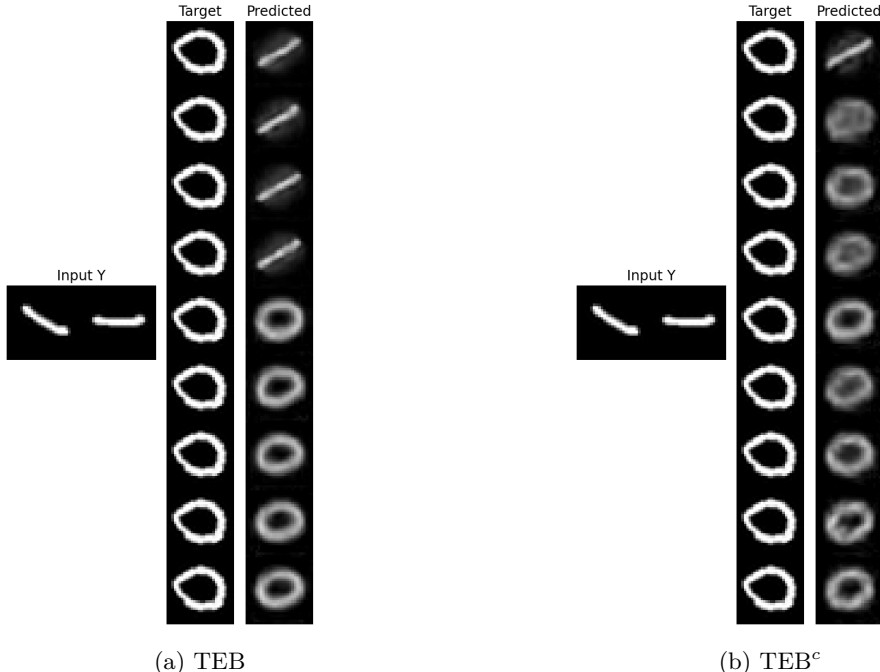

(a) TEB                                   (b) TEB$^c$

Figure 5: Sample testing generation on the same input for varying $\beta$ equal to $0.1, 0.12, 0.14, 0.16, 0.18, 0.2, 0.3, 0.4, 0.5$, from top to bottom respectively. For $\beta = 0.1$ both models learn the trivial representation which does not incorporate class information from the $X$ stream, i.e. change of digit class from 1 to 0. **(a):** The TEB model's representation remains trivial until the sharp increase in reconstruction loglikelihood at $\beta = 0.18$, as seen in Figure 4. **(b):** The TEB$^c$ model's representation appears to transmit class information for $\beta$ as low as 0.12; This is consistent with the appearance (see Figure 4) of the reconstruction loglikelihood dropping off more gradually than TEB for $\beta \geq 0.13$, with the transmitted information very near (distance $< 0.2$ nats) to the true expected transfer entropy for $\beta = .12$ on some seeds.

For the TEB$^c$ models, we pre-trained the context embedding module, which we will call the "$Y$ module", to represent $C$ with next step prediction of $Y'$ using CEB with $\gamma = 1$ on the same dataset. In this case for maximum training efficiency and difficulty for TEB$^c$, we also fix the decoder from the $Y$ module and use it for the reconstruction from $Z$ in the full TEB$^c$ model. We find in Figure 4b that the bifurcation point $\beta_1$ for TEB$^c$ is similar to that recovered for TEB trained end to end. However, we do find that unlike the situation for TEB, here the CMNI point does not appear to be reachable since our bound for $I(Y'; Z|Y)$ continues to increase for $\beta > \beta_1$. On the other hand, it is difficult to find $\beta_0$ directly from Figure 4b, but as shown in Figure 5b, TEB$^c$ fails to predict the change of digit class at $\beta = 0.1$, which puts its $\beta_0$ approximately at 0.1.

One reason the CMNI point is not reachable could be due to our representation of the output loglikelihood; As is the case for any gradients proportional to $\ell^2$ or $\ell^1$, the output probability space assumes independence of each pixel, and therefore does not accurately represent the true image probability space for the data. This cause is evidenced by the fact that the generation quality does not perceptibly vary much between models for $\beta > \beta_1$ even though the loglikelihood does vary significantly. Another cause could be that $I(Y'; C)$ was

---

[2] One can see in Figure 4 that our reconstruction loglikelihood bound for $I(Y'; Z|Y)$ (which differs from $I(Y'; Z|Y)$ by a constant) is approximately maximized at $\beta_1$, and since $I(Z; X|Y) \approx I(Y'; X|Y)$ at $\beta_1$, we are close to the CMNI point.

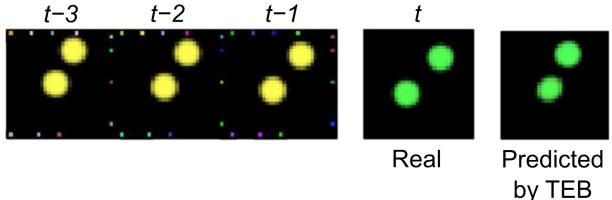

Figure 6: Sample TEB testing sequence of the needle in a haystack task with 20 distractors, with a color switch at the prediction step. Images are arranged from left to right as: three frames of input source stream, next step, generated prediction. For visual clarity, we show only the needle pixel channel group of the source stream, with the first 10 distractor pixels added and randomly placed on the perimeter (note that needle pixel always at top left). The actual streams in the experiments had stacked RGB channel groups, where each group has either a needle or distractor pixel at top left.

not maximized enough due to regularisation. Also, since the $Y$ module does not have any information to reconstruct a randomly swapped in digit, and is never able to represent this on its own, it may not have learned a representation of angle which is disentangled from the identity of the digit. This would result in requiring a higher $\beta$ for learning a $\text{TEB}^c$ model which adjusts the latent by sending extra information from $X$ on sequences which switch, in addition to the class of the digit. To partially test these causes, we check the loglikelihood performance of $\text{TEB}^c$ with a higher $\beta = 10$, from a pre-trained $Y$ module such that it is trained with a higher $\gamma = 100$ to prioritise maximizing $I(Y'; C)$. This results in a significantly improved testing reconstruction loglikelihood of $19.9317 \pm 0.3105$ (as compared to $-16.0291 \pm 0.6041$ at $\beta = 0.2$). Another way to find a better $C$ is to let $Y$ be pre-trained on rotating digits that never switch, because then the training of $Y$ can focus on learning just the rotating of digits. See Figure 3b for an example testing generation in this case with $\beta = 1$. This also demonstrates that TEB is flexible in modulating a pre-trained model for a new task modality.

## 3.2 Detecting a needle in a haystack

Here we design a challenging task to test performance of TEB, and to test its added ability to generalize when the statistical coupling between the $X$ and $Y$ streams is high, but with a conditionally small and crucial directed information transfer from $X$ to $Y$. The task is next step prediction for a colored bouncing balls (Sutskever et al., 2008) video, where the color of the balls can change to one of seven colors at the moment of prediction with probability 0.5. The twist is that there are some number of randomly colored noisy pixels in the video, of which one pixel, which we shall call the "needle pixel", noisily corresponds to the color of the balls in the next step; To predict the next step color correctly, a model has to decide on the three previous frames of video which pixel corresponds to the needle pixel. The sequences for both $X$ and $Y$ are defined as video sequences consisting of the images of the balls and the distractor pixels, but importantly, only $X$ contains the added needle pixel. To make the task of predicting color more difficult still, we introduce a baseline intensity in all color channels (red, green, and blue), at each pixel of each ball. This means that the output reconstruction component of the loss, which is common to all models (for deterministic models it is the only component of the loss), depends only mildly on predicting the correct color. As such, $X$ and $Y$ are very strongly correlated, both with each other and over time. Moreover, the transfer entropy provided by the needle pixel is crucial to this color prediction task, though this information is relatively very small. This ensures that our task fits the description of a large correlation of the sequences, but a small and crucial directed information transfer from $X$ to $Y$. See Appendix B.3 for more detailed descriptions on the utilized $X$ and $Y$ video sequences with the distractor pixels.

In our experiments, we also trained three baselines for comparison. The first two are deterministic baselines. Between them, the first is simply a deterministic version of the TEB model, with the same architecture and latent size, but we dropped the sampling from $q(z|x, y)$ and excluded the KL divergence term in the objective, which is the first term in the TEB objective function. It uses directly the mean of $q(z|x, y)$ as the input for the decoder. The second which we refer to as the joint stream deterministic model or "Deterministic joint" in Table 1 and 2, is a ResNet18 (He et al., 2016) + LSTM which takes all the $X$ and $Y$ information

as a unified input, followed by the same deterministic decoder architecture as TEB. Note that particularly in this task, every image of $Y$ is included in $X$, thus $X$ itself is a valid unified input, and we adopted it in our implementations. The third baseline is CEB, with the same architecture as the $Y$ module of TEB, with added backward encoder. As with "Deterministic joint", CEB takes a unified input as well. The reasons behind including joint stream models, i.e. "Deterministic joint" and CEB, as comparisons are to showcase the advantage when bottlenecking transfer entropy in this task.

To evaluate performance of the directed information transfer from $X$ to $Y$, we want to determine whether the predicted image at time $t$ has the correct color for the balls. To quantify the color accuracy we use an additional color classification network. This classification network is trained to classify color into one of seven color classes, on images of the balls from the training set. Since this is an easy task, the classification achieves 100% accuracy on the test set as well. As such, any errors in color classification reflect errors in the color of the predicted frame, and so, the color classification serves as a metric of color prediction, i.e. the transferred information.

Table 1 shows that on the color prediction task for 20 distractors, the deterministic version of the TEB architecture does not perform much better than what one achieves by simply predicting a constant color as previous frames, which gives an accuracy of approximately 57%. In contrast, TEB model picks up a transfer entropy signal via information bottleneck, which allows it to predict the color at a much better accuracy. TEB outperforms the deterministic baseline consistently across different number of distractors, with an increasing gap for the harder task. In comparison, both the joint stream deterministic and stochastic models (CEB) do not even learn to generalize better than the constant color baseline on 5 distractors. One explanation is that they are likely dealing with a harder task than TEB and "Deterministic", because they cannot make use of the difference from X to Y in the latent space, which is essentially the information denoting the needle pixel. Another explanation lies in the joint stream formulation. For example for CEB, it has to compress the joint stream input as a whole, including the bouncing balls images, needle pixel and all the distractors, so it might ignore the needle pixel which is relatively small.

Table 1: Testing color prediction accuracy on the needle in a haystack task (8 seeds), best performance is shown in bold.

| Model | 5 distractors | 10 distractors | 15 distractors | 20 distractors |
|---|---|---|---|---|
| TEB | **94.64 ± 2.21 %** | **88.39 ± 3.63 %** | **81.99 ± 3.04 %** | **68.37 ± 3.92 %** |
| Deterministic | 93.87 ± 0.94 % | 88.33 ± 2.57 % | 73.27 ± 7.50 % | 59.97 ± 7.26 % |
| Deterministic joint | 38.35 ± 0.75 % | - | - | - |
| CEB | 40.78 ± 0.67 % | - | - | - |

Table 2: Testing reconstruction log-likelihood on the needle in a haystack task (8 seeds), best performance is shown in bold.

| Model | 5 distractors | 10 distractors | 15 distractors | 20 distractors |
|---|---|---|---|---|
| TEB | -551 ± 25 | -568 ± 26 | -562 ± 17 | -566 ± 17 |
| Deterministic | **-515 ± 26** | **-542 ± 21** | **-561 ± 25** | **-558 ± 28** |
| Deterministic joint | -573 ± 30 | - | - | - |
| CEB | -823 ± 66 | - | - | - |

However, Table 2 shows that unlike the case of color prediction accuracy, the aggregate statistics of reconstruction loglikelihood do not differentiate between any model except for CEB, which trades loglikelihood for better identifying the transfer entropy than "Deterministic joint", no matter the number of distractors. This is due to the added whitening on each of the color channels at each ball pixel. As such, the reconstruction can still be largely accurate even when the system is not getting the color correct. In other words, the majority of the reconstruction quality is dependent on the localisation of the balls and not the color of

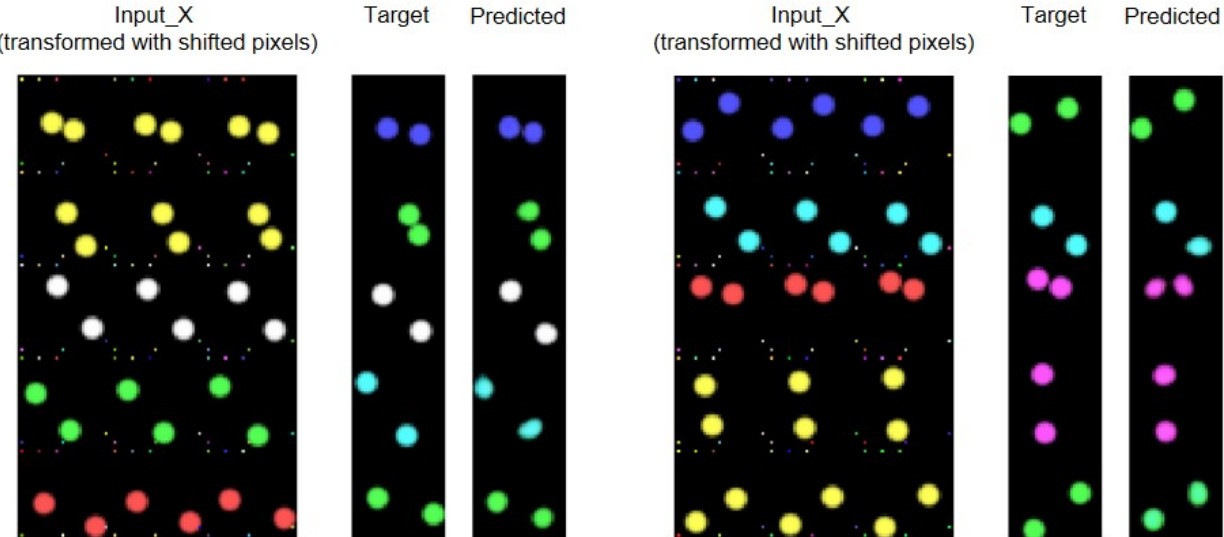

Figure 7: Sample TEB testing generation for 5 distractors

the balls, which highlights the high difficulty disentangling the needle pixel signal, and obtaining a quality representation which allows for the prediction of the true color.

Lastly, we show batch sample generations of TEB for 5 distractors in Figure 7. We believe that it can even be a challenging task for people, without knowing that the needle pixels are fixed always on the top left corner, and the models would have to infer the needle pixels based on the history. See Appendix B.3 for more sample generations in both training and testing and the common generation error made by TEB.

### 3.3 Extrapolating time-series

We also evaluate the performance of TEB on synthetic time-series data, where each time-series signal is generated as the average of five sinusoidal waves of distinct frequencies $f \in \{0.2, 0.4, 0.6, 0.8, 1.0\}$Hz with starting point randomly in $[-1, 1]$. Each sinusoidal wave is generated by unrolling a 2-dimensional continuous time dynamical system with purely imaginary eigenvalues, and recording the $y$ coordinates of the resulting trajectories. Note that with initial conditions lying on the unit circle, the trajectories of the said dynamical system are rotations on the unit circle. See Appendix B.4 for more detailed descriptions on the implemented dynamical system. Note that this is done to facilitate the sudden change of frequency at any given time-step. We refer to this task as multi-component sinusoids. As with other tasks discussed above, a switch with probability 0.5 is also implemented in this dataset, where the frequency of one random component is resampled every time a switch occurs. The task is to extrapolate the future sequence of length 20 for a given time-series signal of length 100. Here $Y$ is the averaged signal and $X$ is the noisy frequencies for all the sub-components for the next time-step. The difficulty of this task comes in largely from the averaging nature of $Y$, as there might be multiple ways to decompose the same signal. Hence it requires the model to capture the information embedded in the transfer entropy in order to understand the true decomposition based on the frequencies.

For this task, we compare TEB model with three other baselines. The first is "Deterministic" defined above, that is simply a deterministic version of the TEB model architecture. The other two are both joint stream models, taking the concatenation of $X$ and $Y$ as a unified input. Between them, the second is simply a deterministic LSTM, which outputs the next sequence as a vector, whereas the third is similar to the latentODE model introduced in Chen et al. (2018), with an LSTM encoder and a neuralODE decoder, but here we only ask the decoder to predict the extrapolation points, instead of reconstructing the inputs as the

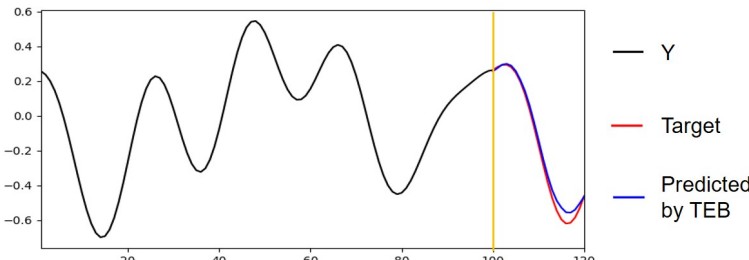

Figure 8: Sample TEB testing sequence of the multi-component sinusoids task when there is a switch at prediction time-step 100, indicated by the yellow vertical line. Note that the sharp change of curve at the prediction point is due to a larger span in the change of frequency of the fourth sub-component, from 0.8 to 0.2.

Table 3: Testing reconstruction log-likelihood on the multi-component sinusoids task (3 seeds), best performance is shown in bold.

| Model | on the entire set | only when switching |
|---|---|---|
| TEB | **4.3497 ± 0.0761** | **3.6839 ± 0.1973** |
| Deterministic | 4.2279 ± 0.0499 | 3.4123 ± 0.1264 |
| LSTM | 3.9941 ± 0.0046 | 2.8577 ± 0.0168 |
| latentODE-VIB | 3.7902 ± 0.0020 | 2.5475 ± 0.0146 |

original latentODE. As with the original version, it uses a fixed Gaussian as a prior, so it is functionally close to VIB, and we refer to it as latentODE-VIB.

Table 3 shows the reconstruction log-likelihood both on the entire testing set and only when switching to a new frequency. Note that since we allow switching to the same frequency, the second scenario comprises around 40% of the entire testing set. Judging from the results, TEB outperforms all the baselines in both cases. In addition, although all the algorithms suffer from a reduction in performance when evaluating only when there is change of frequency, TEB is able to achieve the smallest reduction among the four. This indicates that TEB does a better job capturing the transfer entropy in this task than all other baselines. Furthermore, we show batch sample generations of TEB in Figure 9, where rows 3 in the left and 2 in the right show a common extrapolating error made by TEB. See Appendix B.4 for a case of large deviation in TEB's testing generation, which is more rarely seen in our experiments.

## 4 Discussion and future work

In this paper we developed TEB, an information bottleneck approach designed to capture the transfer entropy between a source and target stream of data. In doing so, TEB brings the benefits of IB approaches to dual stream modeling problems where disentangling statistical dependencies can be very challenging. As well, TEB allows one to use pre-existing encoding models for the target stream. We showed experimentally that TEB allows one to arrive at high quality latent representations that obey the CMNI principle, and use these to make more accurate predictions about the target stream even when the total information transferred is small relative to the joint information.

A limitation of TEB, transpiring from Theorems 1 and 2, is that TEB enables an exact estimation of the true transfer entropy only at convergence to the optimum of the loss for a particular $\beta$, which is the $\beta$ and optimum reaching the CMNI point. Recall that $\beta$ is a weighting coefficient in TEB's loss function, where a larger $\beta$ focuses more on the generations of the outputs and a smaller $\beta$ focuses more on reducing the KL divergence. In practice depending on the architecture, learning, and dataset, the optimum may not be reached and CMNI learning may not be possible. Additionally, if CMNI learning is possible, the identification of this particular $\beta$ may not be possible. However, as shown in the rotating MNIST experiment, we identify

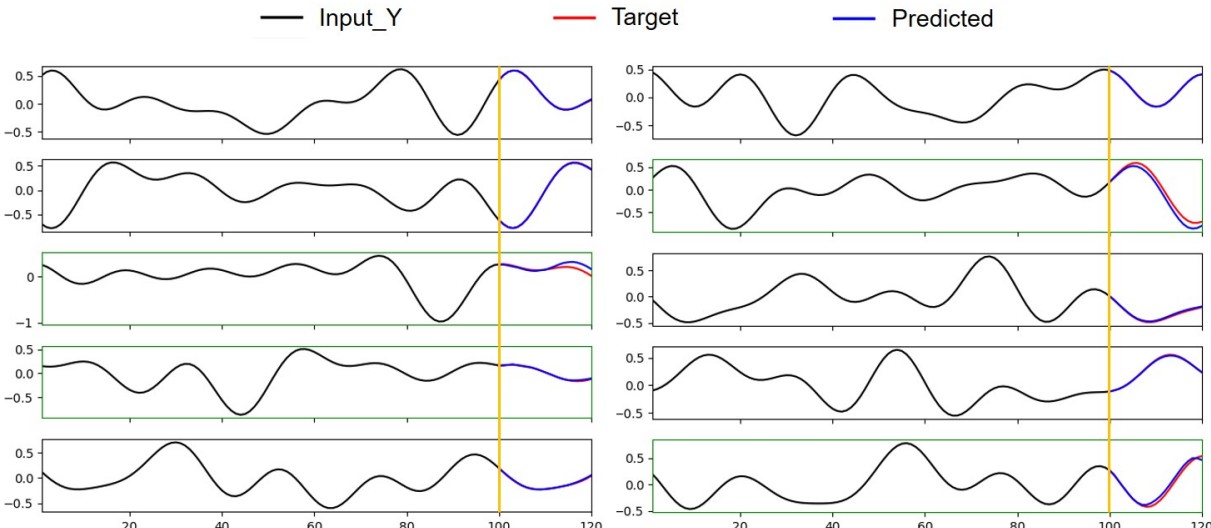

Figure 9: Sample TEB testing time-series sequence generation when there is a switch at prediction time-step 100, indicated by the yellow vertical line. The samples with a frequency switch are marked with green frames.

a phenomenon similar to the one observed for the usual IB case (Wu et al., 2019), which can provide for the approximate identification of this $\beta$. We expect this to occur in some form for most datasets when learning can reach a CMNI point optimum. But, more importantly, estimating the transfer entropy is not the primary purpose of TEB, so even when one cannot get a hold of an estimate on this quantity, TEB can still use the transfer entropy to improve its predictions.

TEB has many potential applications. In principle, TEB could be used in any situation where one possesses multimodal sequential data with statistical dependencies. This could include videos, natural language captioning, financial modeling, medical device time-series, and more. In all cases, an appropriate architecture would have to be built, but TEB can be applied generally. In fact, TEB can potentially be applied to traditionally deterministic architectures for dual data streams tasks by adding some stochasticity to the representations like the technique of Achille & Soatto (2018). As well, TEB could be modified to work in situations with more than two streams of data. Future work will examine the utility of TEB with real data in these potential applications and new architectures, and potential theoretical guarantees of learnability as done in Wu et al. (2019) for the IB loss. This current work provides the analytical foundations of TEB and shows with synthetic data that it works well in principle.

## Broader Impact Statement

This work provides a method for prediction from dual-stream data sources. As with other machine learning approaches to prediction, there may be questionable uses that such a system could be applied to. To avoid such problematic situations, use of TEB should be carefully considered by researchers in advance, and any potential harms should be considered before applying TEB to a new dataset.

## Acknowledgments

This work was done in partnership with BIOS Health Ltd. (https://www.bios.health/) and supported by MEDTEQ+ (Impact Grant 13-F Neuromodulation), Healthy Brains, Healthy Lives (Neuro-Partnerships Program: 3d-13), Mitacs (Accelerate Program: IT19551), NSERC (Discovery Grant: RGPIN-2020-05105; Discovery Accelerator Supplement: RGPAS-2020-00031), and CIFAR (Canada AI Chairs to BAR and GL; Learning in Machine and Brains Fellowship to BAR). GL further acknowledges the Canada Research Chair in Neural Computations and Interfacing (CIHR). This research was also enabled in part by support provided by (Calcul Québec) (https://www.calculquebec.ca/en/) and Compute Canada (www.computecanada.ca).

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

# A  Mathematical formulation

This section is written in a standalone succinct format for maximum clarity on the mathematical details. We presume the reader has some familiarity with information bottleneck methods.

## A.1  Setup

First we state the definition of transfer entropy:

**Definition 1.** Given sequences of random variables $(X_i), (Y_i)$, the **transfer entropy** from $(X_i)$ to $(Y_i)$ at $t > 0$ (with horizon $0 < \ell \leq \infty$) is the conditional mutual information:

$$
\begin{aligned}
T_{X \xrightarrow{t} Y} &= I(Y_t; (X_i)_{t_\ell \leq i < t} | (Y_i)_{t_\ell \leq i < t}) \\
&= H(Y_t | (Y_i)_{t_\ell \leq i < t}) - H(Y_t | (Y_i)_{t_\ell \leq i < t}, (X_i)_{t_\ell \leq i < t})
\end{aligned}
$$

where $t_\ell = \max(t - \ell, 0)$.

Note that the conditional mutual information between two random variables conditioned on another, is just a special case of transfer entropy, and so all the methods developed apply to that case.

Latent variable encoder-decoder models seek to learn a conditional density $p(y|x)$ by learning an encoded latent representation $Z$ of $X$, with an encoder $q(z|x)$ which factorises the true density as a Markov chain with $Z$ independent of $Y$ given $X$ (i.e. $Z \leftarrow X \leftrightarrow Y$):

$$
p(y, z, x) = p(y, x)p(z|x) = p(y, x)q(z|x), \tag{7}
$$

where $p(z|x) = q(z|x)$ since we introduce $Z$. As well, we use a decoder $d(y|z)$ that approximates $p(y|z)$. The functional Markov structure that comes from feed-forward computing $z$ from $x$, and $y$ from $z$, implicitly assumes that the distribution is also factorisable as:

$$
p(y, z, x) = p(x)p(y|z)p(z|x) \approx d(y|z)q(z|x)p(x). \tag{8}
$$

Our situation is analogous to this, but in our setting we actually wish to learn a latent encoding $Z_t$ of $(X_i)_{i<t}$ via a map $q$, which is conditional on $(Y_i)_{i<t}$, from which one is able to decode $Y_t$ conditionally on $(Y_i)_{i<t}$ with the decoder $d$. Put another way, we want to be able to predict $Y$ at time $t$ using the values of $Y$ and $X$ up to time $t - 1$, but using our encoding of $X$ and $Y$ via $Z$. Namely, denoting:

$$
Z = Z_t, \; X = (X_i)_{t_\ell \leq i < t}, \; Y = (Y_i)_{t_\ell \leq i < t}, \; Y' = Y_t, \tag{9}
$$

Assuming the conditional independence $Y' \perp Z | X, Y$., we have $Z, d, q$ are such that:

$$
p(y', z, x, y) = p(y', x, y)p(z|x, y) = p(y', x, y)q(z|x, y). \tag{10}
$$

as illustrated in Figure 10a.

Furthermore, we require the distribution to be learned by a feed-forward encoder-decoder structure, which assumes the following decomposition of joint probability distribution (as illustrated in Figure 10b, equivalent to the conditional independence $Y' \perp X | Z, Y$.), where $d(y'|z, y)$ is a variational approximation to $p(y'|z, y)$:

$$
p(y', z, x, y) = p(x, y)p(y'|z, y)p(z|x, y) \approx p(x, y)d(y'|z, y)q(z|x, y). \tag{11}
$$

We call the method we develop using this structure the **Transfer Entropy Bottleneck (TEB) algorithm**.

*Remark* 1. As an aside, we note that for the TEB method we will also want to consider the case where there is an existing model for encoding $Y$, $q_Y(c|y)$ which could be any existing model for encoding $Y$ with a variable $C$ (e.g. a pre-trained self-supervised model). Thus, $Y$ is first encoded into its own latent $C$, which will not be optimised with the derived optimisation procedure (see Figures 10c and 10d); So in reading, keep in mind that in the text one can replace "$Y$" with "$C$", and every result still holds.

For the rest of the text assume $t$ is fixed and we will use the notations defined in Equation 9 above.

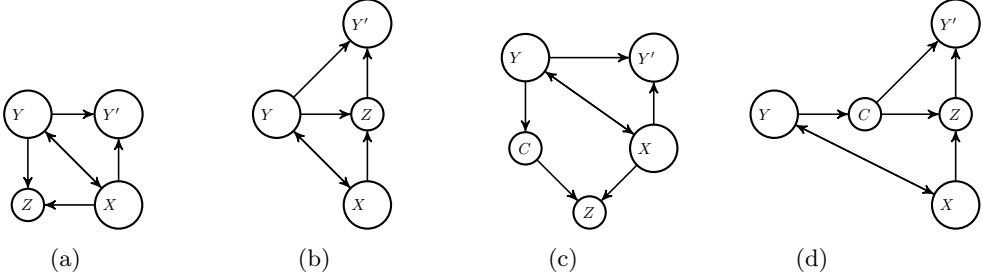

|  (a)  |  (b)  |  (c)  |  (d)  |

Figure 10: Probabilistic graph representing the random variables and their relationships as assumed by TEB. **(a)**: The actual random variables in the dataset. Independencies include $Y' \perp Z|X, Y$. **(b)**: The random variables as computed by our feed-forward computation. Independencies include $Y' \perp X|Z, Y$. **(c)**, **(d)**: The actual and feed-forward graphs, respectively, when utilising a pre-trained context encoding $C$. Independencies for (c) include $Y' \perp Z|X, C$, $Y' \perp C|Y$, and $Y' \perp C|X, Y$, and for (d) include $Y' \perp X|Z, C$ and $Y' \perp Y|X, C$.

As is standard in information bottleneck methods, we follow a minimum necessary information principle, in which we wish to learn our latent representation $Z$ such that it captures exactly the necessary conditional information (conditioned on $Y$) between $X$ and $Y'$.

Specifically we would like the following to hold, which, in analogy to the usual minimum necessary information principle, we call the Conditional Minimum Necessary Information point or CMNI point:

$$I(Y'; X|Y) = I(Y'; Z|Y) = I(Z; X|Y) \tag{CMNI}$$

Due to the probabilistic structure of our setting, we do not have $I(Y'; Z|Y) \leq I(Y'; X|Y)$ and $I(Y'; X|Y) \leq I(Z; X|Y)$ directly from the data processing inequalities, as opposed to the standard IB setting (where $I(Y; Z) \leq I(Y; X)$ and $I(Y; X) \leq I(Z; X)$). Thus, we first need to prove that we can arrive at the CMNI point by the following procedure:

1. maximizing $I(Y'; Z|Y)$        (CMNI procedure)
2. minimizing $I(Z; X|Y)$

To prove this, first we remind the reader of the chain rule for conditional mutual information here since we will use it throughout the text:

*Recall.* The chain rule for conditional mutual information is

$$I(A; B|C) = I(A; B, C) - I(A; B)$$

or

$$I(A; B, C) = I(A; B) + I(A; B|C).$$

We can then prove the following proposition:

**Proposition 3.** *The following inequalities hold under the independencies implied by Equation 10 and 11:*

$$I(Y';Z|Y) \leq I(Y';X|Y) \leq I(Z;X|Y).$$

*Proof.* Inequality 1 $(I(Y';Z|Y) \leq I(Y';X|Y))$: By the chain rule

$$
\begin{aligned}
I(Y';X,Y,Z) &= I(Y';Z,Y) + I(Y';X|Z,Y) \\
&= I(Y';X,Y) + I(Y';Z|X,Y)
\end{aligned}
$$

however, Equation 10 implies $Y' \perp Z|X,Y$, and so $I(Y';Z|X,Y) = 0$ which implies

$$I(Y';Z,Y) \leq I(Y';X,Y).$$

Since

$$
\begin{aligned}
I(Y';Z|Y) &= I(Y';Z,Y) - I(Y';Y) \\
I(Y';X|Y) &= I(Y';X,Y) - I(Y';Y),
\end{aligned}
$$

we get that $I(Y';Z|Y) \leq I(Y';X|Y)$.

Inequality 2 $(I(Y';X|Y) \leq I(Z;X|Y))$: By the chain rule:

$$
\begin{aligned}
I(X;Y',Z,Y) &= I(X;Y',Y) + I(X;Z|Y',Y) \\
&= I(X;Z,Y) + I(X;Y'|Z,Y)
\end{aligned}
$$

however, Equation 11 implies $X \perp Y'|Z,Y$, and so $I(X;Y'|Z,Y) = 0$ which implies

$$I(X;Y',Y) \leq I(X;Z,Y).$$

Since

$$
\begin{aligned}
I(X,Z|Y) &= I(X;Z,Y) - I(X;Y) \\
I(X,Y'|Y) &= I(X;Y',Y) - I(X;Y),
\end{aligned}
$$

we get that $I(Y';X|Y) \leq I(Z;X|Y)$. $\qquad\square$

*Remark* 2. Note that in the proof above, if we had used Equation 11 and $X \perp Y'|Z,Y$ in the first part in addition to 10, we would have $I(Y';X|Y) \leq I(Y';Z|Y)$, and therefore $I(Y';X|Y) = I(Y';Z|Y)$. This shows that our feed-forward generation of $Y'$ cannot represent the true distribution unless it maximizes $I(Y';Z|Y)$.

## A.2 A transfer entropy bottleneck objective for CMNI learning

We will now prove that we can use these structures to derive a tractable loss function for optimizing a model in accordance with the CMNI procedure in order to reach the CMNI point. This method is the TEB algorithm.

First, in-line with item 1 of the CMNI procedure, we want to maximize $I(Y';Z|Y)$. One can decompose the information $I(Y';Z|Y)$ as

$$I(Y';Z|Y) = -H(Y'|Z,Y) + H(Y'|Y) \propto -H(Y'|Z,Y). \tag{12}$$

Importantly, the term $H(Y'|Y)$ is fully determined by the true data distribution, and thus, can be dropped from the optimization procedure, which is why the proportionality holds. However, this will lead to the problem that the information $I(Y';Z|Y)$ will not be tractable to calculate from our existing procedure. We will address this in the later section A.3 with another form of the TEB algorithm.

We can readily bound $-H(Y'|Z,Y)$ from below with:

$$
\begin{aligned}
-H(Y'|Z,Y) &= \mathbb{E}_{p(y,z,y')}[\log(p(y'|z,y))] \\
&= \mathbb{E}_{p(z,y)}\left[D_{\mathrm{KL}}\left(p(y'|z,y)\|d(y'|z,y)\right)\right] + \mathbb{E}_{p(y,z,y')}[\log(d(y'|z,y))] \\
&\geq \mathbb{E}_{p(y,z,y')}[\log(d(y'|z,y))]
\end{aligned}
\tag{13}
$$

Thus, in order to maximize $I(Y';Z|Y)$ we can simply maximize $\mathbb{E}_{p(y,z,y')}[\log(d(y'|z,y))]$. Moreover, since maximizing this bound also minimizes the expected KL divergence

$$
\mathbb{E}_{p(y,z,y')}[\log(p(y'|z,y))] - \mathbb{E}_{p(y,z,y')}[\log(d(y'|z,y))],
$$

we will simultaneously also learn to approximate $p(y'|z,y)$ with $d(y'|z,y)$, and the bound will become tight.

Now, per the second item of the CMNI procedure, we want to minimize $I(Z;X|Y)$. To obtain a tractable upper bound for $I(Z;X|Y)$ one needs to do more work, and use more distributional approximations. In VIB, one bounds $I(Z;X)$ above with

$$
I(Z;X) \leq \mathbb{E}_{p(x)}\left[D_{\mathrm{KL}}(q(z|x)\|p(z))\right],
$$

where $p(z)$ is a pre-specified fixed prior. We shall see that $I(Z;X|Y)$ can be bounded above similarly, however our "prior" cannot be a hand-set fixed distribution, since it must be relevant to $Y$. We introduce an additional encoder $q_y(z|y)$ for learning a conditional prior on the latent $Z|Y$ that is informative for the full TEB computation. This gives a clear bound of $I(Z;X|Y)$, which becomes tighter over training and thus can also serve as an estimate of $I(Z;X|Y)$ at convergence (and therefore an estimate of $I(Y';X|Y)$):

$$
\begin{aligned}
I(Z;X|Y) &= \mathbb{E}_{p(y,z,x)}\left[\log\left(\frac{q(z|x,y)}{p(z|y)}\right)\right] \\
&= \mathbb{E}_{p(y,z,x)}\left[\log\left(\frac{q(z|x,y)}{q_y(z|y)}\right)\right] - \mathbb{E}_{p(y)}\left[D_{\mathrm{KL}}(p(z|y)\|q_y(z|y))\right] \\
&\leq \mathbb{E}_{p(y,z,x)}\left[\log\left(\frac{q(z|x,y)}{q_y(z|y)}\right)\right] = \mathbb{E}_{p(x,y)}\left[D_{\mathrm{KL}}(q(z|x,y)\|q_y(z|y))\right].
\end{aligned}
\tag{14}
$$

Training $q_y(z|y)$ by minimizing this bound, also gives $q_y(z|y) \approx p(z|y)$, since it squeezes the KL divergence of the two.

Equation 13 with Equation 14 gives the Lagrangian loss to minimize for the transfer entropy bottleneck algorithm:

**Theorem 4.** *Under the independence assumptions implied by Equation 10 and 11, the minimum of the following objective with respect to $q, q_y, d$ will be at the CMNI point for $Z, X, Y, Y'$, for some hyperparameter $\beta > 0$:*

$$
TEB = \mathbb{E}_{p(x,y)}\left[D_{\mathrm{KL}}(q(z|x,y)\|q_y(z|y))\right] - \beta\mathbb{E}_{p(y,z,y')}[\log(d(y'|z,y))].
\tag{TEB}
$$

### A.3  Transfer entropy bottleneck from a learned context encoding

Suppose one has already learned an encoding $q_y(\cdot|y)$ for $Y$ that provides a latent representation of $Y$. This section will show one can use this encoding in place of $Y$ in the TEB algorithm. This can come with many practical advantages like computational efficiency, training efficiency, reuse of existing models, and rapid prototyping when experimenting with multiple configurations or hyperparameter settings for the full TEB model.

To avoid confusion with the latent space $Z$ of the TEB model, we will refer to this latent encoding for $Y$ as $C$. We integrate the computation of $C$ into the TEB model according to Figures 10c and 10d. As promised by remark 1, here we invoke the fact that if $C$ is fixed when following the procedure (CMNI procedure), so that Equation 12 holds when replacing "$Y$" with "$C$", then all of the results of the previous sections hold when replacing "$Y$" with "$C$". Since we actually care about bottlenecking conditioned on $Y$ rather than $C$, and measuring the information $I(Y';X|Y)$ rather than $I(Y';X|C)$, we need the following lemma

**Lemma 5.** *If $Y' \perp Y|X, C$, $Y' \perp C|X, Y$, $Y' \perp C|Y$, and $q_y(c|y)$ is such that $I(C; Y')$ is maximized, then*

$$I(X; Y'|C) = I(X; Y'|Y).$$

*In particular, one has $I(C; Y') \approx I(Y; Y')$ gives $I(X; Y'|C) \approx I(X; Y'|Y)$.*

*Proof.* Since $Y' \perp C|Y$, the data processing inequality gives $I(C; Y') \leq I(Y; Y')$, and so maximizing $I(C; Y')$ takes us to $I(C; Y') = I(Y; Y')$. Now, expanding $I(Y'; X, Y, C)$ two different ways gives

$$\begin{aligned} I(Y'; X, Y, C) &= I(Y'; X, Y) + I(Y'; C|X, Y) \\ &= I(Y'; X|Y) + I(Y'; Y) + I(Y'; C|X, Y), \end{aligned}$$

and

$$\begin{aligned} I(Y'; X, Y, C) &= I(Y'; X, C) + I(Y'; Y|X, C) \\ &= I(Y'; X|C) + I(Y'; C) + I(Y'; Y|X, C). \end{aligned}$$

Using the facts that $I(C; Y') = I(Y; Y')$, $I(Y'; Y|X, C) = 0$ by $Y' \perp Y|X, C$, and $I(Y'; C|X, Y) = 0$ by $Y' \perp C|X, Y$, we get

$$\begin{aligned} I(Y'; X|Y) + I(Y'; Y) &= I(Y'; X|Y) + I(Y'; Y) + I(Y'; C|X, Y) \\ &= I(Y'; X|C) + I(Y'; C) + I(Y'; Y|X, C) = I(Y'; X|C) + I(Y'; Y), \end{aligned}$$

and therefore $I(Y'; X|Y) = I(Y'; X|C)$. $\qquad\square$

The condition that $I(C; Y') \approx I(Y; Y')$ via $q_y(c|y)$, means that we (approximately) do not lose information about $Y'$ using $C$ in $Y$'s place. As such, $q_y(z|y)$ is a perfectly reasonable prior for the latent $Z|Y$ of the TEB model (which implies that $Z$ is in the same vector space as $C$). Thus, we design the following loss such that we assume $p(z|y) := q_y(z|y)$ by definition. Noting that we required $C$ and thus $q_y$ to be fixed, this allows us to use a new version of the loss function (TEB) with $C, q_y$ as:

**Theorem 6.** *Under the independence assumptions $Y' \perp Z|X, C$, $Y' \perp C|Y$, $Y' \perp C|X, Y$, $Y' \perp X|Z, C$ $Y' \perp Y|X, C$, and $C \sim q_y(\cdot|y)$ such that $I(C, Y') = I(Y, Y')$, the minimum of the following objective with respect to $q, d$ will be at the CMNI point for $Z, X, Y, Y'$, for some hyperparameter $\beta > 0$:*

$$TEB^c = \mathbb{E}_{p(x,y)}\left[D_{\mathrm{KL}}(q(z|x, c)\|q_y(z|y))\right] - \beta\mathbb{E}_{p(y,z,y')}[\log(d(y'|z, c))]. \tag{TEB$^c$}$$

Let us also suppose now that we have a decoder $d_y(y'|c)$. This will allow us to estimate $I(Y'; Z|Y)$ separately, at convergence of the decoders $d, d_y$, even if $\beta$ or $\gamma$ are such that we do not compress $Z$ enough to converge to the CMNI point, which can be seen from the following proposition:

**Proposition 7.** *If the distributions $d(y'|z, c), d_y(y'|c)$ are such that $d(y'|z, c) \approx p(y'|z, c)$ and $d_y(y'|c) \approx p(y'|c)$, then*

$$I(Y'; Z|Y) \approx \mathbb{E}_{p(y',z,c)}\left[\log(d(y'|z, c)) - \log(d_y(y'|c))\right].$$

*Proof.* The proof is immediate from the assumptions, and the fact that

$$I(Y'; Z|Y) \approx I(Y'; Z|C) = \mathbb{E}_{p(y',c,z)}\left[\log(p(y'|z, c)) - \log(p(y'|c))\right],$$

$\qquad\square$

The above approximation is readily interpretable, since it is the expected difference between the log-likelihoods of $Y'$, as estimated by $d$ and $d_y$. In practice, the above estimates are good when one knows the form of the output distribution and thus is able to correctly parameterize it with our approximate distributions. For example, when the outputs are categorical as in a classification task.

*Remark* 3. Finally, although we are assuming the representation $C$ is not trained by the TEB loss and is fixed, this assumption is only needed for the proportionality in Equation 12. If we let $q_y$ be optimized by the TEB loss, but at the same time continue to also optimize $q_y$ such that $I(Y'; C)$ stays maximized during the optimization process, then $H(Y'|C) = -I(Y'; C) + H(Y')$ remains fixed, and so Equation 12 remains valid.

We find in practice that if we let $q_y$ be optimized by the TEB loss, but continue to train $q_y$ to maximize $I(Y'; C)$ in the same way that it was pretrained, $I(Y'; C)$ does indeed stay maximized and the TEB procedure works as well.

## B  Additional implementation and experimental details

### B.1  Additional model implementation details

All of the implementations were done using PyTorch (Paszke et al., 2019). The parameters of the models were initialized using the default He initialization using normal distribution (He et al., 2015). We used Adam optimizer (Kingma & Ba, 2015) with the default parameters; an initial learning rate of $10^{-4}$ with no weight decay. All the experiments were conducted on a single GPU with 48G memory. For all models the latent dimension of 128 was used in all non convolutional hidden layers, including the hidden dimension size of the LSTMs.

The implementation of the TEB model uses two initial encoders for the two streams $q_y$ and $q_x$, both of which feed into two separate LSTMs to aggregate information across the sequence. The encoder depends on the input modality; on images the initial encoders are ResNet18, on a discrete modality like class integers, we used an embedding vector lookup table (i.e. a `torch.nn.Embedding`), and on time-series it is just the identity mapping. The hidden state from the $Y$ pathway LSTM is then linearly projected into a $128 \times 3$ dimensional vector. The first two 128 dimensional chunks correspond to the means $\mu_y$ and log variances $\log(\sigma_y^2)$ for the 128 dimensions of the prior $q_y(z|y)$, which are treated as being independent Gaussians in each dimension. The last 128 dimensional chunk is fed, along with the output of the initial $X$ stream encoding $q_x$, into an MLP with one hidden layer with a ReLU activation function (Nair & Hinton, 2010)[3]. The outputs of this MLP are means $\mu_x$ and log variances $\log(\sigma_x^2)$. These lead to the parametrization of the 128 independent Gaussian dimensions of $q(z|x,y)$ as having means $\mu = \mu_y + \mu_x$, and log variances $\log(\sigma_x^2)$, and the latent representation $Z$ is then sampled from this distribution. We note here that we initialise the last linear layer of the MLP computing $\mu_x$ and $\log(\sigma_x^2)$ so that $\mu_x$ is the zero vector, and $\log(\sigma_x^2)$ is $\log(10^{-7})$ in every coordinate.

For rotating MNIST and needle in the haystack tasks, the decoder $d$ is a positional encoding followed by a convolutional network, which results in an image with dimensions $(c, h, w)$, where $c = 1$ for rotating MNIST and $c = 3$ for the needle in a haystack task. To create the positional encoding, we first arrange a spatial grid of shape $(4, h, w)$ where each channel is the distance to each image border for a given coordinate, normalized into [0,1]. This grid is passed through a learnable linear layer which maps every coordinate into 128 dimensional space, and results in a positional encoding of shape $(128, h, w)$. The sampled $Z$ is copied and repeated in each spatial dimension into a tensor with same shape as the positional encoding, and is then added to the positional encoding. The output image is then obtained by passing this through a space preserving convolutional component of the decoder, where all convolutions have stride 1 and are padded (with reflection padding) to preserve spatial dimensions. For example for rotating MNIST the decoder has the following architecture:

```
(1):  Conv2d(features=128, kernel_size=(7, 7)), BatchNorm, ReLU

(2):  Conv2d(features=64, kernel_size=(5, 5)), BatchNorm, ReLU

(3):  Conv2d(features=32, kernel_size=(5, 5)), BatchNorm, ReLU

(4):  Conv2d(features=16, kernel_size=(5, 5)), BatchNorm, ReLU

(5):  Conv2d(features=8, kernel_size=(3, 3))

(6):  Conv2d(features=4, kernel_size=(3, 3)), BatchNorm, ReLU

(7):  Conv2d(features=c, kernel_size=(3, 3))
```

, whereas for the needle in the haystack task, the decoder has all the layers above except for 5 and 6.

---

[3]For the datasets in our experiments, we found that using an MLP as such was not necessary at all for performance, and we could just use $q_x$ to output $\mu_x, \log(\sigma_x^2)$ directly. However, the MLP may be necessary when using a dataset where the processing of the $X$ stream depends on the particular $Y$ example. Thus we kept the MLP in the architecture as such for full generality.

For multi-component sinusoids task, the decoder $d$ is neuralODE, implemented using the `torchdiffeq` package. The dynamics of the ODE is parameterized by a MLP with one hidden layer with 50 hidden units and a ReLU activation function, and another MLP with the same hidden units is utilized to map each latent state to the scalar output at each time-step. The decoder uses default *dopri5* ODE solver for forward propagation, representing Runge-Kutta of order 5 of Dormand-Prince-Shampine. The ODE solver is tasked to solve an initial value problem for a sequence of time-steps from 0 to 1 second with an interval of 0.05 second. To calculate the gradient, since the memory cost is not our concern, we chose to directly backpropagate through the operations of the solver instead of using the adjoint method, for a reduce in the running time. Lastly, the output sequence of the decoder starts with the last point of the input, as we found it to be beneficial for the overall performance.

To optimize the TEB loss on these architectures we interpret our output as probabilistic, so that we can calculate the output loglikelihood term in the TEB loss. In our interpretation we assume each pixel in each channel is independently distributed with means equal to the output image, and when a variance parameter is required for the distribution we fix the variance to 0.1. This interpretation of the output can give reconstruction loss gradients proportional to common usual loss functions like mean squared error, $\ell^1$, or various others, with the particular loss depending on the chosen output distribution. For rotating MNIST and multi-component sinusoids we choose a Gaussian output distribution, giving reconstruction gradients proportional to mean squared error. For the needle in a haystack task we could have used the same interpretation, however training times were significantly longer than for rotating MNIST, so instead we interpreted each pixel as a binary intensity, as we found this speed up training time without compromising performance. This gives reconstruction loss proportional to a binary cross entropy with logits across pixels.

Also for the needle in a haystack experiments, to test the accuracy of the models in reconstructing the correct color out of the 7 color classes, we trained a color classifier on the training set. The color classifier is a convolutional network with the following architecture:

```
(1):  Conv2d(features=16, kernel_size=(5, 5), stride=1, padding=2), ReLU

(2):  MaxPool2d(kernel_size=(2, 2), stride=2)

(3):  Conv2d(features=32, kernel_size=(3, 3), stride=1, padding=1), ReLU

(4):  MaxPool2d(kernel_size=(2, 2), stride=2)

(5):  Conv2d(features=64, kernel_size=(3, 3), stride=1, padding=1), ReLU

(6):  MaxPool2d(kernel_size=(2, 2), stride=2)

(7):  Linear(output_size=7)
```

## B.2    Rotating MNIST details and samples

The rotating MNIST task was generated from a base dataset of videos of 500 examples of rotating MNIST digits (and an additional 500 validation and 500 testing base videos). In a given base video sequence, a digit starting upright rotates counter-clockwise $\frac{\pi}{4}$ at every frame, for 6 frames. The dataset was generated from this base dataset by sampling 300 possible random digit changes in the last frame of every possible triple of consecutive frames in a base video, with a given switching probability of 0.5. Note that when a digit is switched, the new digit has the same angle as the digit it is replacing. Also, although a new switched digit is from a different example digit, the class of the switched digit may be the same as the previous one.

Note that we can calculate the ground truth transfer entropy for rotating MNIST task as follows. Firstly, note that digits' class labels are the only mutual information captured in $I(Y'; X|Y)$, without loss of generality,

we use $Y'$, $X$ and $Y$ to denote their classes respectively. Note that the task is such that we have

$$p(y'|x,y) = p(x|y,y') = \begin{cases} 1, & \text{if } x = y' \\ 0, & \text{otherwise} \end{cases},$$

and we have equal probability for each digit class

$$p(y) = \frac{1}{\text{size}(Y)} = \frac{1}{10}.$$

We can decompose transfer entropy for rotating MNIST as follows:

$$
\begin{aligned}
I(Y'; X|Y) &= \sum_{x,y,y'} p(y',x,y) \left[ \log \left( \frac{p(y'|x,y)}{p(y'|y)} \right) \right] \\
&= \sum_{x,y,y'} p(y)p(y'|y)p(x|y',y) \left[ \log \left( \frac{p(y'|x,y)}{p(y'|y)} \right) \right] \\
&= - \sum_{y',y \; : \; y'=x} p(y)p(y'|y) \left[ \log \left( p(y'|y) \right) \right] \\
&= - \sum_{y',y} p(y)p(y'|y) \left[ \log \left( p(y'|y) \right) \right] \\
&= - \frac{\text{size}(Y)}{\text{size}(Y)} \sum_{y'} p(y'|y=k) \left[ \log \left( p(y'|y=k) \right) \right] \\
&= - \sum_{y'} p(y'|y=k) \left[ \log \left( p(y'|y=k) \right) \right]
\end{aligned}
\tag{15}
$$

where $k$ is any of the digit classes. Following the equation, the ground truth transfer entropy is 1.67689 nats, with

$$p(y'=k'|y=k) = \begin{cases} 0.55, & \text{if } k' = k \\ 0.05, & \text{otherwise} \end{cases}.$$

We arrange each input sample for the $Y$ module $q_y$ as a sequence of 2 input frames of a rotating digit preceding a possible switch. The input for the $X$ module $q_x$ is a length 2 sequence of the class of the digit in the next frame. The target $Y'$ is the next frame of the video where a possible digit switch may have occurred.

For these experiments we train the TEB models on 3 seeds, for every $\beta$ from 0.1 to 1 incremented by tenths, with extra hundredths increments in the chaotic regime in $[.1, .2]$. End to end TEB models were trained for 10 epochs whereas $\text{TEB}^c$ models were trained for 5 epochs. The $\text{TEB}^c$ models used a $Y$ module trained for 20 epochs to represent $C$ with next step prediction of $Y'$ using CEB with $\gamma = 1$. In this case for maximum training efficiency and difficulty for $\text{TEB}^c$, we also fixed the decoder from the $Y$ module and use it for the reconstruction from $Z$ in the full $\text{TEB}^c$ model.

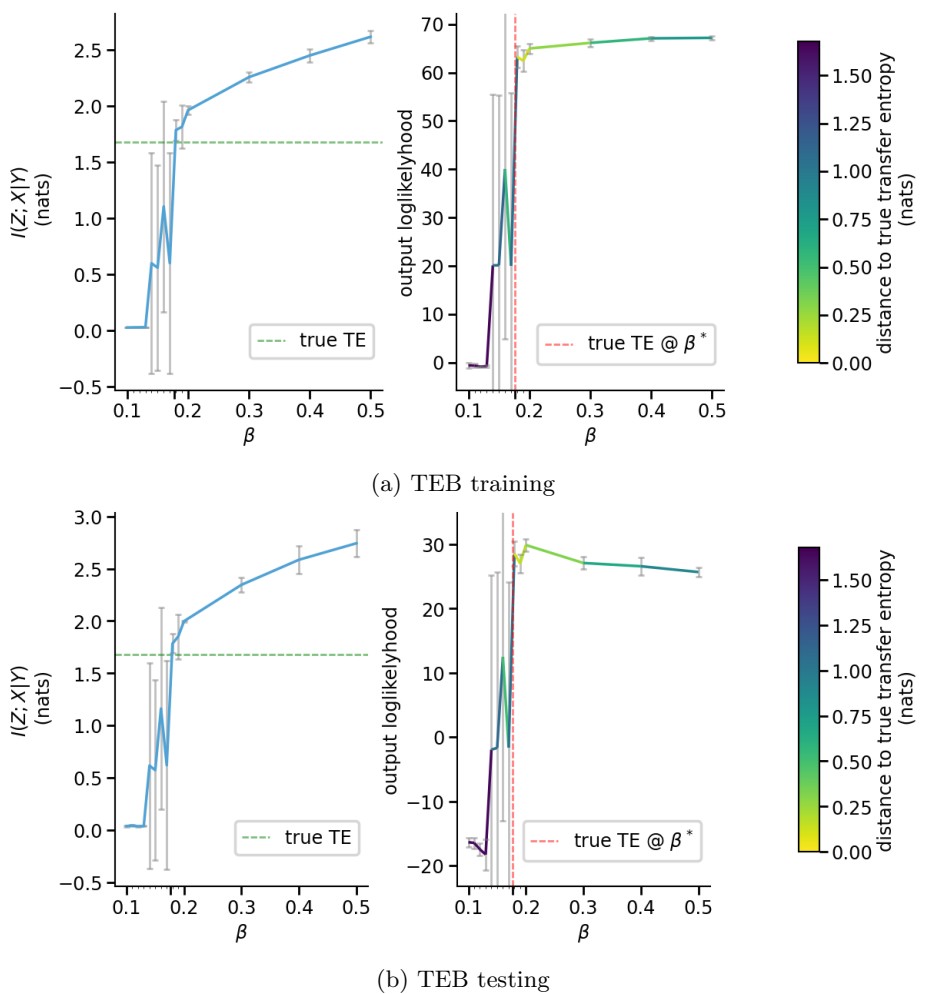

(a) TEB training

(b) TEB testing

Figure 11: Plots of the information metric and reconstruction loglikelihood vs $\beta$ incremented by tenths, with extra hundredths increments in the chaotic regime in $[.1, .2]$, for TEB testing. The green dashed horizontal line in the left subfigure indicates the true transfer entropy for this dataset, from which we obtain $\beta^*$ as the specific value of $\beta$ achieving the true transfer entropy, calculated as the interpolated value of $\beta$ for which $I(Z; X|Y) = I(Y'; X|Y)$ on average from repeated trials. In the right subfigure, the output loglikelihood of the model trained with the obtain $\beta^*$ is marked using the red dashed vertical line. Note that the variance of runs highly increases for $\beta \in (.1, .2)$, but the models in this interval that found a solution with $I(Z; X|Y) \approx I(Y'; X|Y)$ are the same models that achieved higher reconstruction performance.

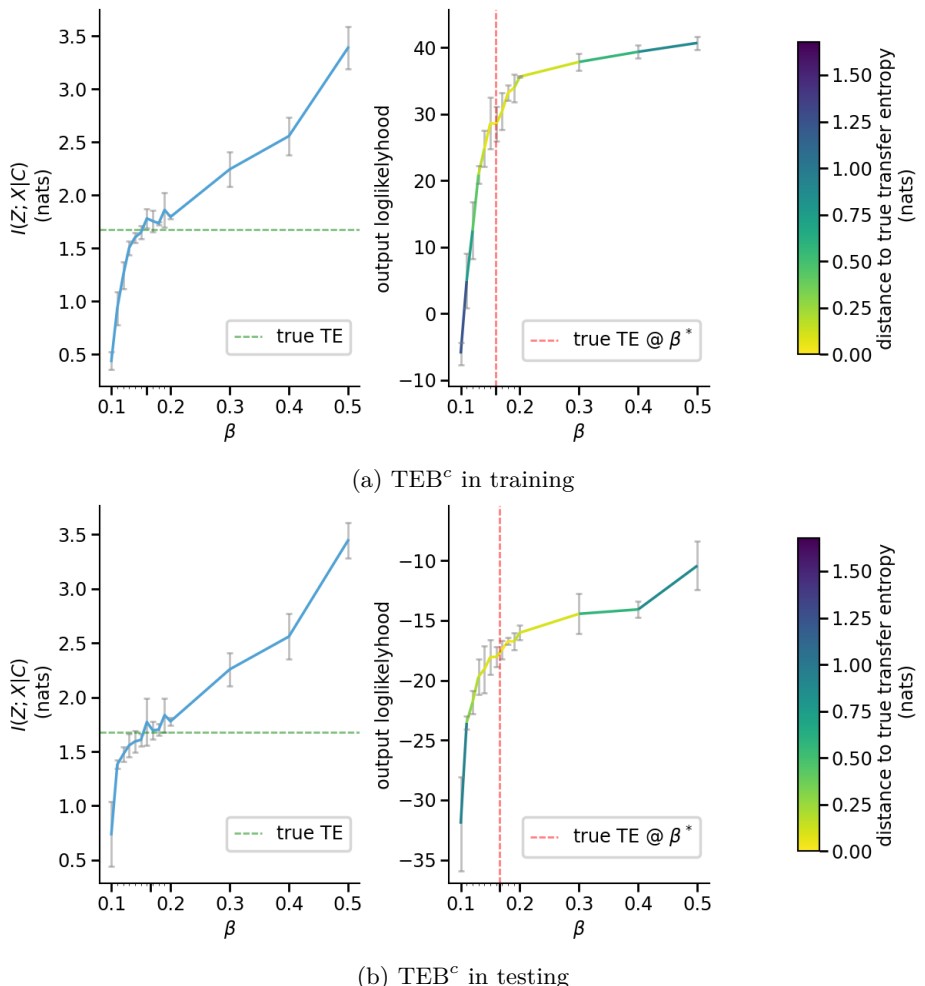

(a) TEB$^c$ in training

(b) TEB$^c$ in testing

Figure 12: Same plots as Figure 11, but for TEB$^c$

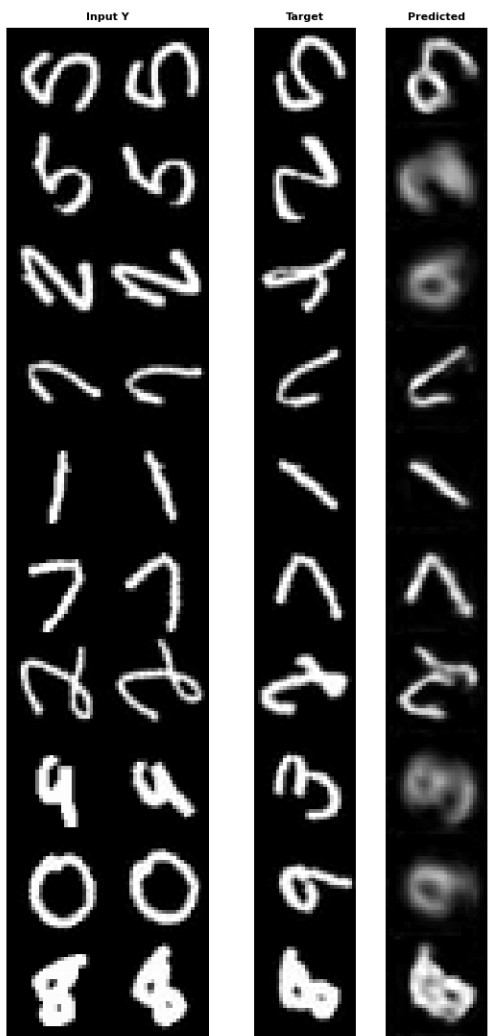

Figure 13: Sample TEB testing generation with $\beta = 0.2$.

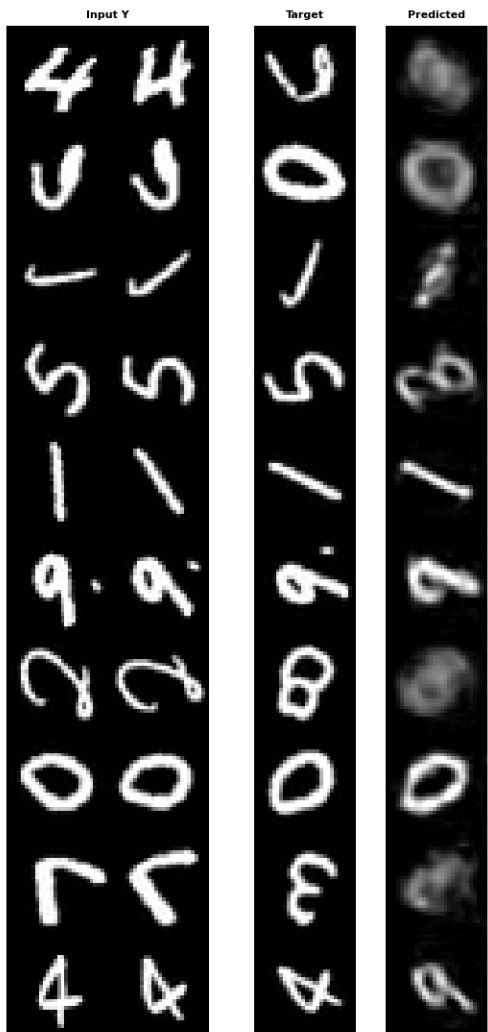

Figure 14: Sample TEB$^c$ testing generation with $\beta = 0.2$. Quality is somewhat worse in comparison to TEB. For discussion relevant to this, we refer the reader to the end of section 3.1, and also Figure 15.

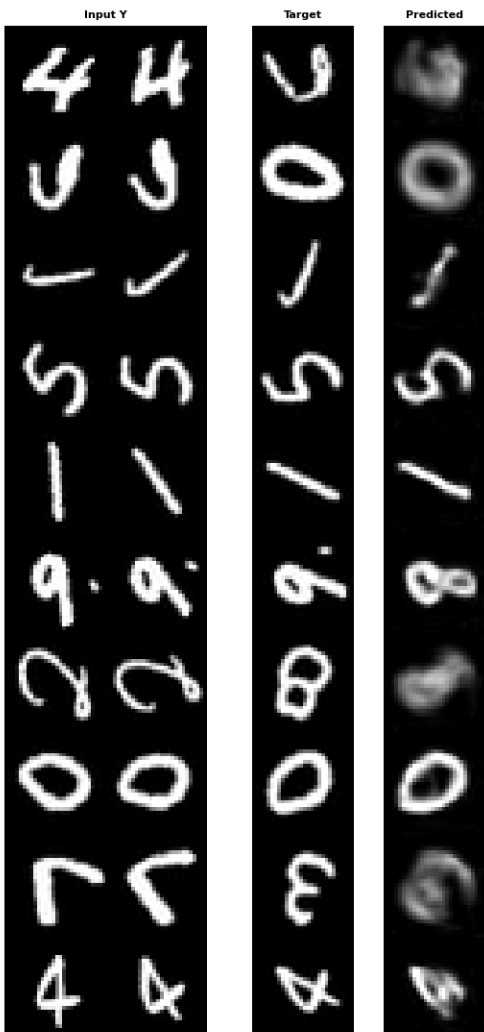

Figure 15: Sample TEB$^c$ testing generation with $\beta = 10$, which helps speed up learning, and with a pre-trained $Y$ module which maximizes $I(Y'; C)$ more than in Figure 14 and 12; $Y$ module is trained with the CEB method with $\gamma = 100$, as opposed to $\gamma = 1$ for Figure 14. Note the cleaner generation in comparison to Figure 14, especially when the $Y$ module does the majority of the "work" in the case where digits do not switch.

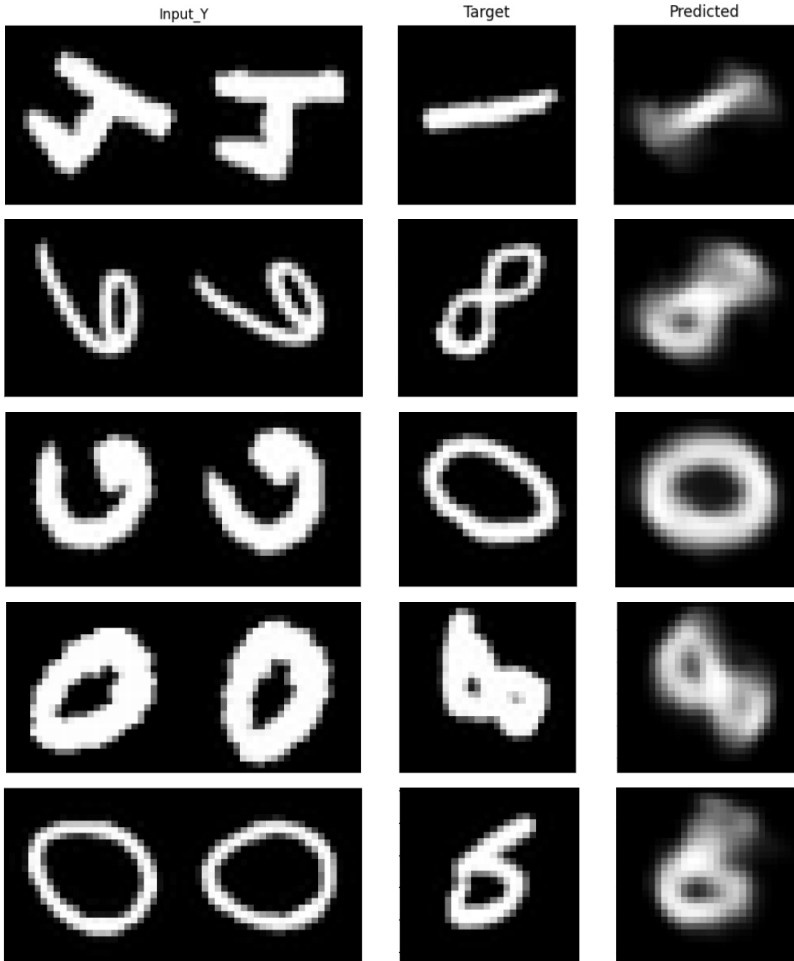

Figure 16: Sample TEB$^c$ testing generation for $\beta = 1$, trained on digits which always switch, and where the $Y$ module is pre-trained on rotating digits that never switch. This demonstrates the flexibility of TEB in modulating a pre-trained model with a new modality. This TEB model was trained for 30 epochs as opposed to 5 for all other TEB$^c$ experiments (or 10 for all other TEB experiments), which improves reconstruction quality and consistency.

### B.3 Needle in a haystack details and samples

The needle in a haystack task base dataset consists of 142 videos of two colored bouncing balls consisting of 6 frames, in each of 7 distinct color classes (and an additional 75 validation, and 75 testing, base videos in each color class). The chosen color classes are all possible RGB triples with values in the alphabet $\{85, 255\}$, except $(85, 85, 85)$. The balls all have a minimum intensity of 85 in all red, green, blue color channels so as to bound the distance between images of different colors with the same ball configuration, thus adding difficulty.

The dataset was generated from this base dataset by sampling 5 possible random ball color changes in the last frame of every possible 4-tuple of consecutive frames in a base video, where a new ball color class in the last frame was sampled with probability 0.5. In addition to this, $D$ distractor pixels (we generated datasets for all of $D \in \{5, 10, 15, 20\}$ distractors), and one true or "needle" pixel, was generated and incorporated for each such 4 frame video section according to the described procedure as follows.

First the color of each distractor pixel $v \in \{0, 255\}^3$ is randomly (uniformly) chosen each frame from the available 7 color classes. Second, the needle pixel's color in each frame is set to the color class of the balls

in the next frame. Note that both distractor and needle pixels have no minimum baseline brightening (so intensity 85 is replaced with 0). Third, for each pixel including the needle pixel, $u_1, u_2, u_3 \overset{i.i.d}{\sim} U(0, 102)$ noise is sampled, and that pixel's color $(v_i)_{i \leq 3}$ is replaced with $(|v_i - u_i|)_{i \leq 3}$.

To incorporate these $D + 1$ extra pixels, each frame of the 3 initial bouncing ball frames is repeated $D + 1$ times, the repetitions are concatenated in the channel dimension, and the $D + 1$ individual colored pixels are added to the top left of the frame; One pixel is added in each of the $D + 1$ repeated RGB channel groups in a random order.

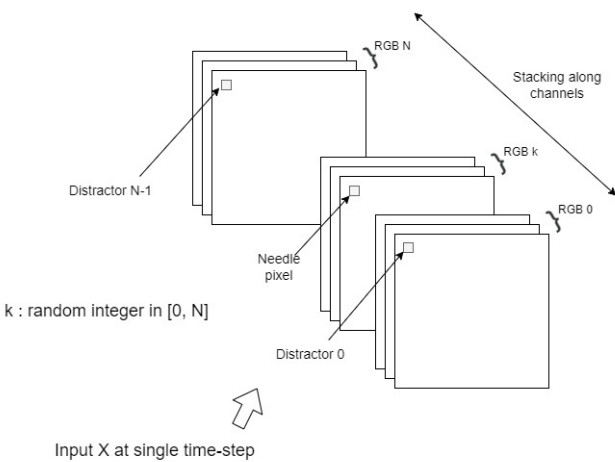

Figure 17: Illustration of repetitions and stackings of the RGB channel groups for input $X$ and $Y$ in the needle in the haystack task.

We arrange each input sample for the $X$ module $q_x$ as such a sequence of 3 input frames. The input for the $Y$ module $q_y$ is this same sequence, but with the RGB channel group containing the needle pixel removed. The target $Y'$ is the next frame of the video of the bouncing balls, where a possible color class switch may have occurred.

We trained all models on 8 seeds and, in each seed, tested the model with the best color prediction accuracy of the color classifier on the validation set. The TEB model was trained for 450 epochs on each dataset. We found the deterministic models stabilised early (Though as we see, the deterministic models are overfitting to the distractor pixels for larger numbers of distractors); Validation and training metrics stabilised well within 300 epochs for 5 distractors (usually within 200 epochs), and well within 400 for all other number of distractors (usually well within 300 epochs), and so we stopped training for deterministic models at 300 epochs for 5 distractors, and 400 epochs for all other distractor numbers.

For the TEB model, we conducted a hyperparameter search for values of $\beta$ between 1 and $10^{12}$, and found that since the task difficulty varied greatly with the number of distractors, different values for $\beta$ could be best for the different datasets. Also, training difficulty meant that the best validating values of $\beta$ were quite high relative to the rotating MNIST experiment. For example, for 20 distractors we found that values of $\beta$ between $10^3$ and $10^5$ achieve the best validation performance. To learn a more bottlenecked representation we also gradually decreased $\beta$ during training from the selected hyperparameter.

To generate the results of Table 1 and 2 with the number of distractors $D \in \{5, 15, 20\}$, we used an initial $\beta = 10^5$, decreasing to $10^4$ at epoch 300, and further decreasing to $10^3$ at epoch 400. For $D = 10$ we used an initial $\beta = 10^3$, decreased to $10^2$ at epoch 350, and decreased further to 10 at epoch 400.

For the CEB model on 5 distractors task, We also conducted a hyperparameter search for $\gamma$ between 0.5 and 100 and found that $\gamma = 10$ achieved the best validating performance.

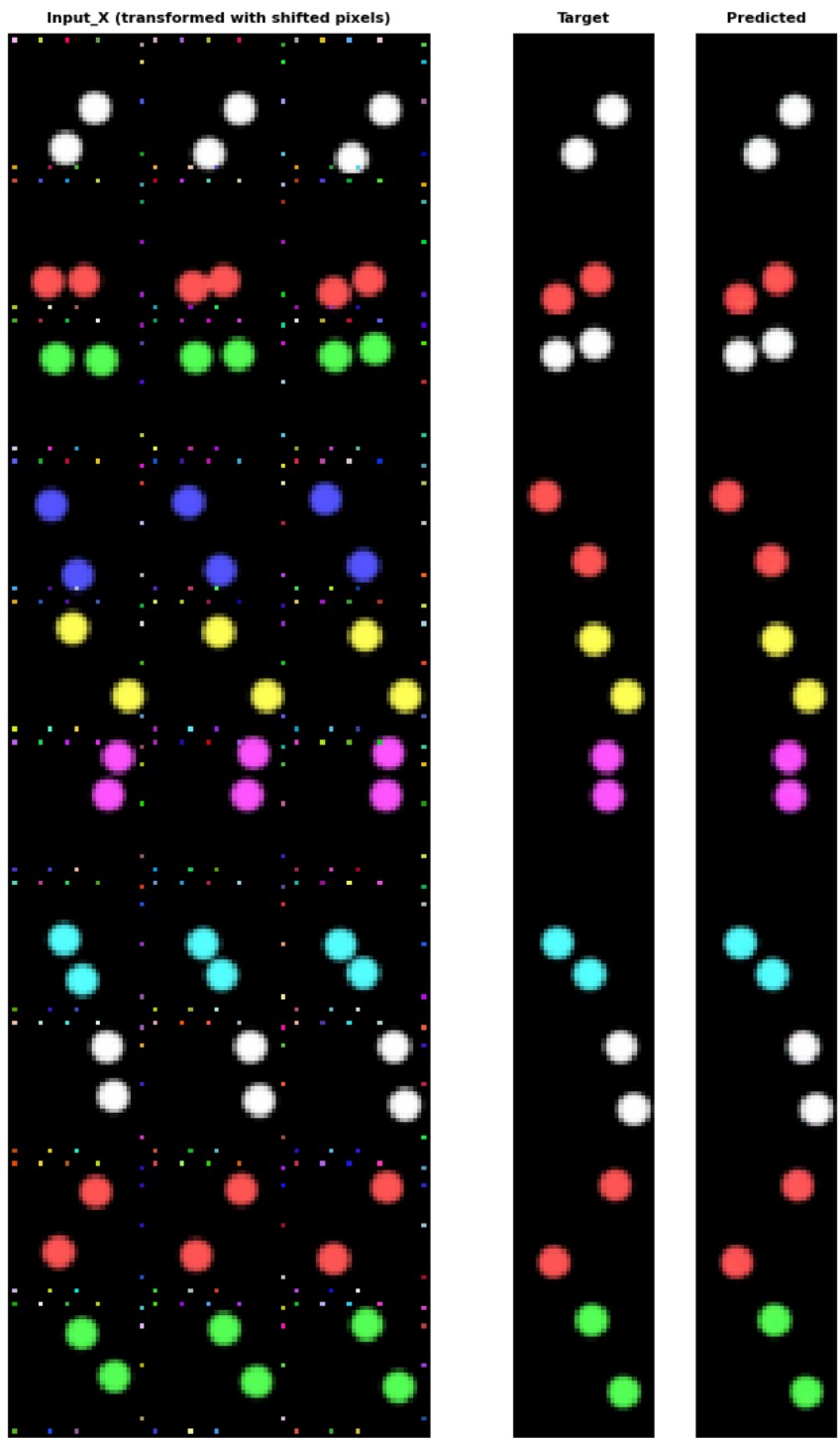

Figure 18: Sample TEB training generation for 20 distractors

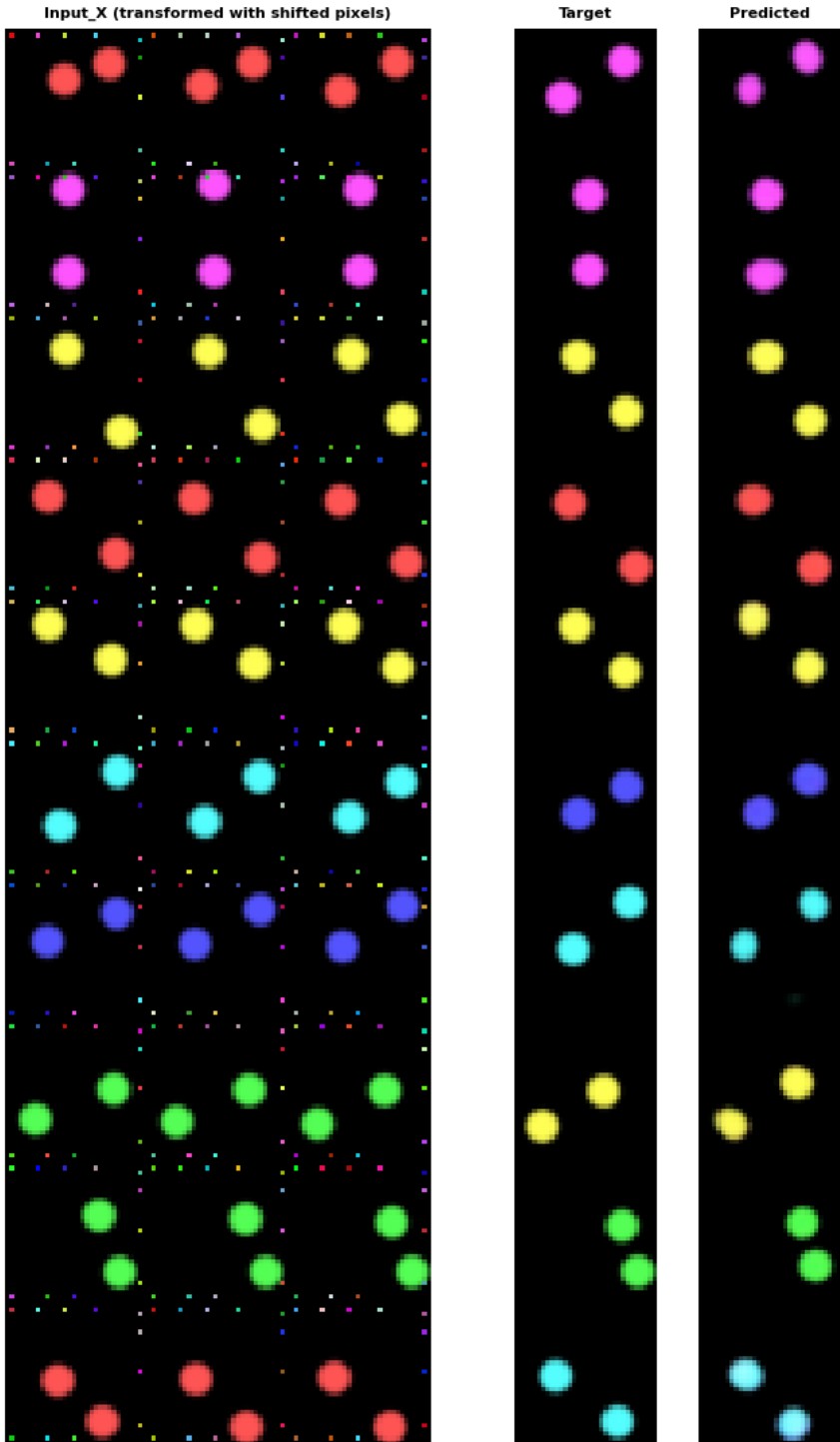

Figure 19: Sample TEB testing generation for 20 distractors

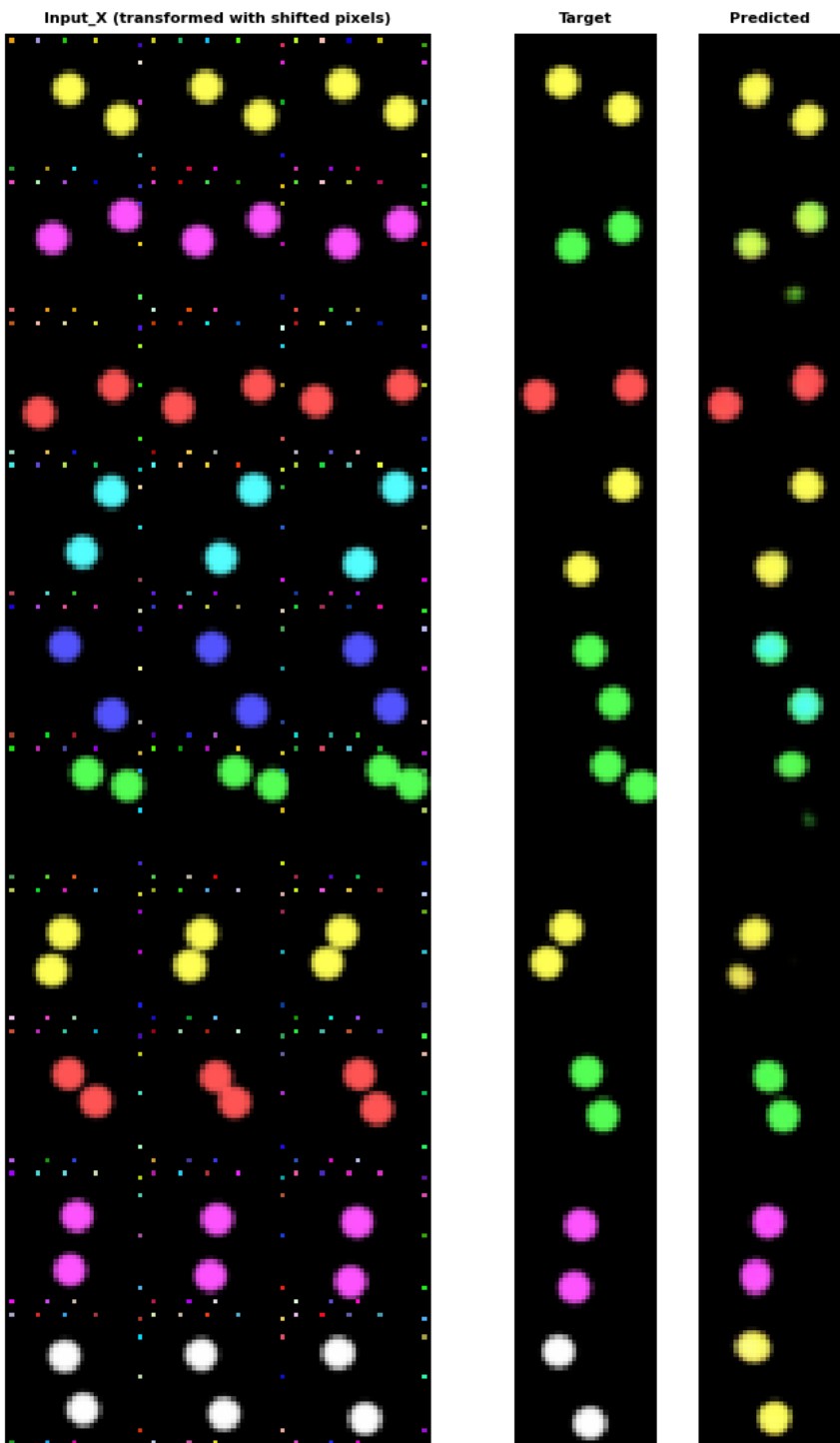

Figure 20: Sample TEB testing generation for 20 distractors, showing prediction errors. At times the color prediction is missed, or balls generated are uneven/irregular.

Lastly, to demonstrate the performance of TEB$^c$ on this challenging dataset, we trained CEB with $\gamma = 1$ on a separate dataset of 5 distractors with the bouncing balls colors never change for 1000 epochs, and used it as the pre-trained Y module to be incorporated into the TEB$^c$ formulation. This is in essence a

simpler task than the normal needle in the haystack task, and the objective of the pre-trained Y module is to predict the localisation of the balls in the target frame while keeping the same color as in the histories. We trained both TEB$^c$ with $\beta = 10$ and "deterministic" with the same pre-trained Y module fixed, on the 5 distractors dataset with 0.5 probability of switching the color, but both models suffered severe overfitting quickly. Therefore we increased the size of the dataset by 20 times, and trained both models on it instead for 8 seeds. We found the testing color accuracy of TEB$^c$ is $93.48 \pm 3.93$ %, while that of the deterministic baseline is $86.13 \pm 4.67$ %. Therefore, introducing an information bottleneck is even more advantageous with a fixed pre-trained Y module, compared to those without as reported in Table 1. One potential reason for the deterministic baseline's poorer performance is that the pre-trained Y module might just not be good enough for it. Alternative reason could be that it is still easier to overfit in this scenario, since here the deterministic model has all the information in the X stream just to find the color changes, without any form of regularizations. This demonstrates that TEB has the capability of incorporating a pre-trained model in this challenging task. However, training TEB$^c$ is considered as a more difficult task than TEB.

### B.4 Multi-component sinusoids details and samples

The multi-component sinusoids task dataset consists of 30k, 5k, and 5k time-series signal of 6 seconds, for training, validation, and testing, respectively. The sampling rate is 20Hz, corresponding to in total 120 points per signal. Each time-series signal has 5 sinusoidal waves with distinct frequencies $f \in \{0.2, 0.4, 0.6, 0.8, 1.0\}$Hz and starting point randomly sampled in $[-1, 1]$ as sub-components, and the order of the 5 components is also random. At the end of the 5th seconds, with 0.5 probability, the frequency of a randomly chosen component will be randomly resampled into one of the 5 frequencies. The objective of the task is to capture the change in frequency and accurately predict the last 20 points of extrapolations given the past.

Each sinusoidal wave of the time-series signal is generated by unrolling a 2-dimensional continuous time linear autonomous dynamical system with purely imaginary eigenvalues.

$$\frac{dy}{dt} = \begin{bmatrix} 0 & 2\pi f \\ -2\pi f & 0 \end{bmatrix} y \tag{16}$$

With initial condition $y_0$ lying on the unit circle, the trajectory of $y(t)$ is clockwise rotation on the unit circle, and we used the the second dimension of the trajectory's Cartesian coordinates as the generated sinusoidal wave for frequency $f$. When there is a switch in frequency, the future trajectory will be unrolled according to the updated dynamical system. The dataset was generated by unrolling the dynamical system using *dopri5* ODE solver of package `torchdiffeq`. The frequencies of all sinusoidal waves were recorded during the dataset generation, and to add difficulty, we added $U(-0.15, 0.15)$ noise to the recorded frequencies post-generation.

We arranged each input sample for the $Y$ module $q_y$ as a length 100 averaged signals of the 5 sinusoidal waves preceding a possible switch. The input for the $X$ module $q_x$ as a length 100 sequence of the frequencies of all waves in the next time-step. The target $Y'$ is the length 21 averaged signal, including the length 20 sequence after a possible frequency switch may have occurred, as well the last time-step of $Y$.

We trained all models on 3 seeds and, in each seed, tested the model with the best reconstruction loglikelihood on the validation set. All models were trained for 2000 epochs. We found that most of the runs reached best validation performance within 500 epochs, after which, the validation performance of TEB and deterministic model stabilised well, while that of latent ODE and deterministic LSTM degraded due to more severe overfitting.

For the TEB and latentODE-VIB, we conducted hyperparameter search for values of $\beta$ between 10 and 1000 and between 10 and 100, respectively, and we found that 1000 for TEB and 50 for latentODE-VIB achieved the best validating performance. In addition, similar as in the needle in the haystack task, with the initial $\beta = 1000$, we adopted the schedule of gradually decreasing $\beta$ by 100 every 100 epochs starting epoch 500, until it reached $\beta = 200$ at epoch 1200. However, unlike the cases in the needle in the haystack task, the schedule was less effective, and we only found that it yielded an improving validating performance (only by a small margin) in one run. Therefore, the schedule might not be necessary for this task.

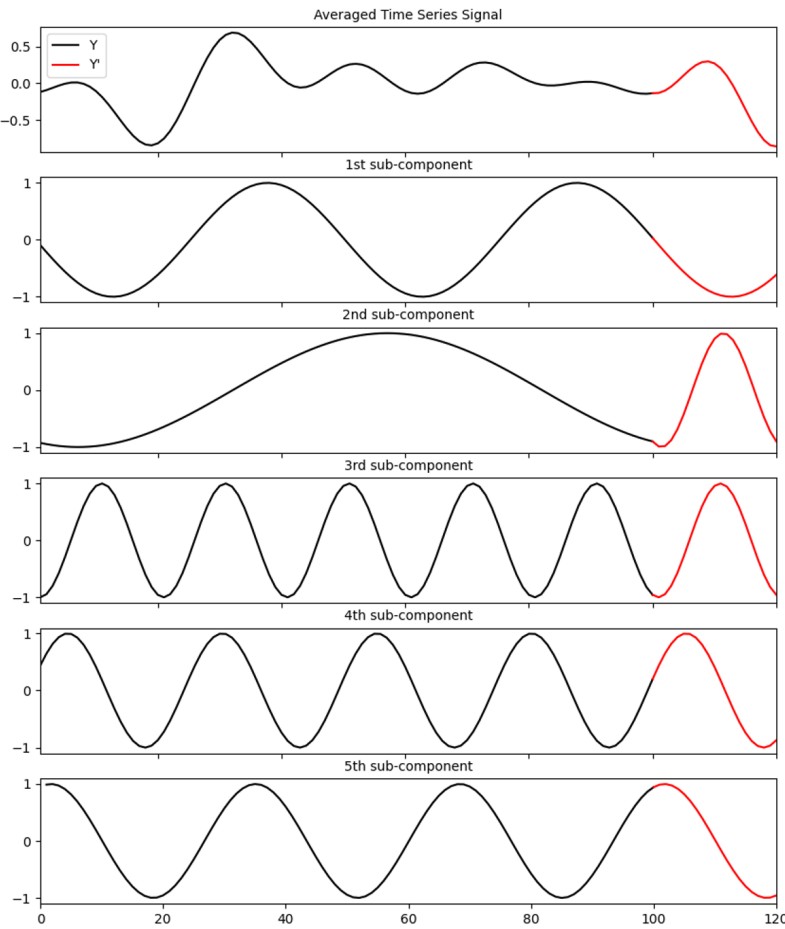

Figure 21: Sample data in multi-component sinusoids dataset. The first row shows the signal used for training in this task, made from averaging all sub-components from the other rows. Shown in the subsequent rows, the sub-components are sinusoidal waves with distinct frequencies, with 0.5 probability to switch frequency at time-step 100. This example demonstrates a switch in the second sub-component from 0.2Hz to 1Hz.

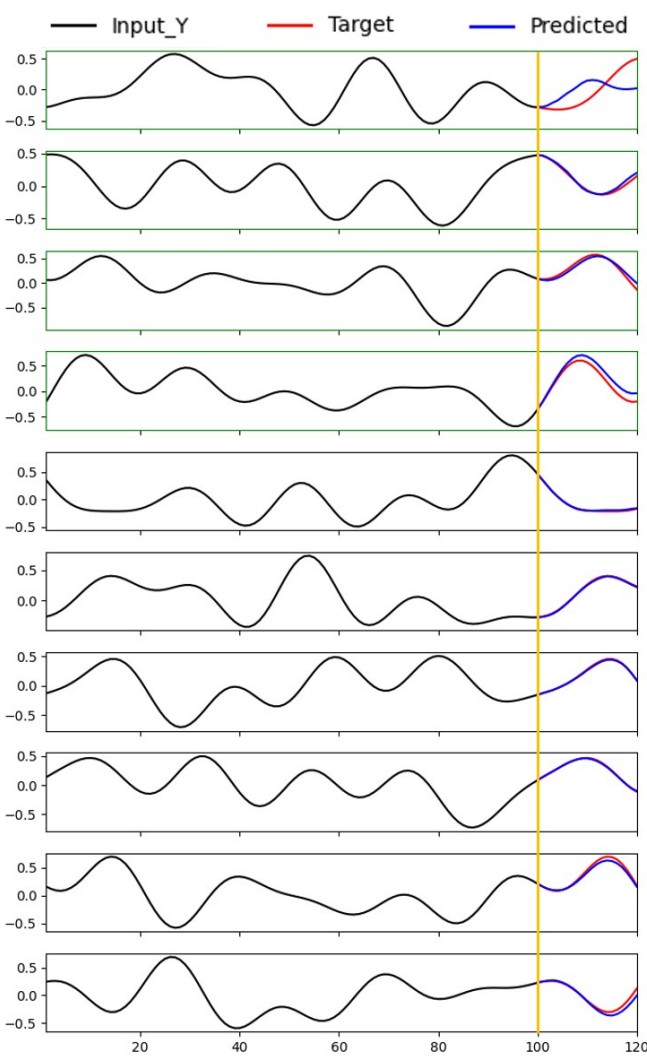

Figure 22: Sample TEB testing time-series sequence generation when there is switch at prediction time-step 100, in which the switches are shown in green frame, showing prediction error. Note the large prediction deviation from the target in the first row. This is more rarely seen case when there is error in predictions, more frequently, the error occurs similar to the 4th row, with accumulated error at later time-steps.

## C  Discussion on TE estimation vs TEB

Here we provide a discussion contrasting methods which estimate transfer entropy with the goals of our TEB method.

The main goal of our paper is learning a transfer entropy bottlenecked representation for prediction, and we provide an estimate of the information transfer of the model only when converged to CMNI point in rotating MNIST task. On the other hand, there are existing methods which focus on the estimation of conditional mutual information and transfer entropy (Ursino et al., 2020; De La Pava Panche et al., 2019; Zhang et al., 2019). Generally, these methods are algorithms which estimate by: kernel density estimation methods, targeting a Donsker Varadhan bound, or by a mutual information estimation approach to separately estimate each component $I(A; B, C)$, $I(A; C)$, of $I(A; B|C) = I(A; B, C) - I(A; C)$.

In principle the bounds for $I(Z; X|Y)$ or $I(Y'; Z|Y)$ we derive may be substituted in our algorithm by any estimator for TE, though as the goal of this paper is to introduce a variational learning technique for learning a transfer entropy bottlenecked representation, examining the combination of these methods is beyond the scope of this paper. In practice the price paid for using a separate learned information estimator to bottleneck the representation are the extra parameters needed for the neural estimation of the information, and a potentially higher variance or less tractable optimisation procedure. For example, Zhang et al. (2019) showed that utilizing directly MINE (Belghazi et al., 2018) estimator for transfer entropy would introduce more variance to the loss. In addition, ITENE algorithm proposed in (Zhang et al., 2019) maximizes and minimizes different MINE estimators for each of the chain-rule-decomposed parts of the conditional mutual information, and one of these parts will not coincide with the direction of optimization of the TEB, and more generally, conditional IB procedure. For example for $I(Z; X|Y) = I(Z; X, Y) - I(Z; Y)$, ITENE estimates $I(Z; X|Y)$ via maximizing MINE estimator of $I(Z; Y)$ and minimizing that of $I(Z; X, Y)$, but IB would require the resulted $I(Z; X|Y)$ estimator to be minimized as a whole. Therefore one would follow a more complicated optimization procedure which alternates maximizing the statistics network parameters with respect to the information estimate, and the minimizing the bottleneck parameters with respect to the information estimate. This could increase the variance of learning, and may require more steps of optimization for the ITENE phase early in learning, to ensure the information estimate provides a meaningful learning signal for the bottleneck.

