# OpenReview forum: "Transfer Entropy Bottleneck: Learning Sequence to Sequence Information Transfer"
_TMLR — Accepted by TMLR_

### Review · Reviewer_7gXH · 2022-12-31

**Summary Of Contributions:**

The paper presents an information bottleneck method for supervised learning when the input can be separated into two sets of variables X and Y (e.g. X and Y are both images, or image sequences), such that most of the useful information is contained in Y but some key information must be extracted from X. Ideas from previous work applying information bottleneck methods to deep neural networks are adapted to this particular setting. The result is a loss function (the TEB loss) for training a supervised learning model, and an approximation for the amount of information extracted from X. Experiments on synthetic data demonstrate the advantage of this approach over standard supervised learning.

**Audience:**

Yes

**Claims And Evidence:**

Yes

**Requested Changes:**

Critical:
- The logic is backwards in Equation 2, which makes the assumption that Y' is independent of X conditioned on Z,Y. This is an assumption in Eq. 2, rather than an implication of it.

Just suggestions:
- In Appendix B.1, the encoder models use ReLU output units which are initialized to zeros. I would suggest using Softmax output units to avoid problems where the gradients are multiplied by zero when backpropagated through the activation.
- Appendix A replicates the main text, almost word-for-word in parts. Fig 2 and 10 and the captions are the same.

**Strengths And Weaknesses:**

Strengths:
- This is an example of using an information theoretic objective function to translate domain knowledge of the data into model inductive biases. This is very appealing and could have wide applications.
- Practically, I like how the same TEB objective is used to optimize all model parameters. The paper doesn't discuss this, but an alternative for learning q(z|x,y) is to use an adversarial model that attempts to predict y from z, such that the two models are optimizing different objectives, resulting in training instabilities. This approach seems to avoid those problems.

Weaknesses:
- Some of the writing could be more clear, including the motivation, assumptions, approximations, bounds, comparisons of the models in experiment 3.2, and the description of Figure 4.
- The motivation behind this method is not completely clear. The paper provides two motivations: (1) Small amounts of crucial information might be ignored in the input; (2) we may want to improve an existing supervised model given extra inputs without retraining. Regarding (1), the synthetic datasets used in the experiments appear to exemplify the applications the authors have in mind, but it would be helpful to have a concise description of where this method should be applied. Reason (2) seems like an afterthought, and the TEB method isn't compared to simpler alternatives for this use case.
- It's not clear to me why the focus is on sequence data. Can't this method be applied to any supervised learning task where the inputs can be divided into a set of primary variables Y and secondary variables X, where X and Y are correlated but X might contain some key information? This seems to be a more general case that would be simpler to consider (especially since the experiments are on synthetic data anyway).
- All experiments are on synthetic data sets that exhibit particular properties that the TEB aims to exploit.

---

> ### Author Response · Authors · 2023-01-27
> **Response to Reviewer 7gXH (1/4)**
>
> We thank the reviewer for the detailed review on our manuscript, please see our point-by-point response as below, please note that we highlight the text changes in the revision in bold:
>
>
> >The logic is backwards in Equation 2, which makes the assumption that Y' is independent of X conditioned on Z,Y. This is an assumption in Eq. 2, rather than an implication of it.
>
> We thank the reviewer for catching the unclear wordings in our paper, and we have updated accordingly to emphasize the assumption we make in Equation (2), and that Equation (2) is equivalent to the conditional independence:
>
> “Furthermore, we require the distribution to be learned by a feed-forward encoder-decoder structure\, **which assumes the following decomposition of joint probability distribution (as illustrated in Figure 2b), where d(y'|z,y) is a variational approximation to p(y'|z,y)\:**”
>
> as well as
>
> “**The first part of Equation 2 is equivalent to the conditional independence Y ′ ⊥ X|Z, Y\.**”
>
> The related sentence for Equation (1) is updated accordingly as well.
>
>
> >In Appendix B.1, the encoder models use ReLU output units which are initialized to zeros. I would suggest using Softmax output units to avoid problems where the gradients are multiplied by zero when backpropagated through the activation. - Appendix A replicates the main text, almost word-for-word in parts. Fig 2 and 10 and the captions are the same.
>
> For the MLP at the end of the encoder, we use ReLU as the activation function of the hidden layer, and its output layer is not associated with a non-linear activation function. This is because the expected outputs of the MLP are mean and log variance, both of which have ranges in the set of real numbers. That being said, it is correct to say that in our implementation, we initialized the MLP of the X stream encoder so that its mean is zero at the beginning. Indeed, we agree with the reviewer that by doing it this way the backpropagation would be blocked at first, but this is in fact what we intend to see. By doing that, we want to impose an implicit constraint that the information from the X stream is just conditional upon the information from the Y stream. In other words, to make sure that when training TEB, the Y encoder is trained before the X encoder. In addition, in our implementation, since the mean of Z fed to the decoder is the sum of the mean from both X and Y encoders, setting only the mean of the X encoder to zero is not likely to result in a zero input to the decoder. We share the reviewer’s concern on zero gradient, hence we empirically examined the gradient during training and found that the zero gradient occurred only at the first backward pass for the X encoder, as expected. We found this architecture performed well in practice.
>
> We intend for Appendix A to be a standalone section for maximum clarity on the mathematical details, so it is expected that the certain content of Appendix A overlaps with the main body of the paper, in terms of text and figure. The choice is made since we believe that it would be helpful to be able to follow through the entire derivations without interruptions.
>
>
> >Some of the writing could be more clear, including the motivation, assumptions, approximations, bounds, comparisons of the models in experiment 3.2, and the description of Figure 4.
>
> We apologize for the insufficient clearness in our paper. We have updated the paper in the “introduction’’ and “‘methods” section accordingly, as described below in our responses, as well as descriptions on model comparison in the needle in the haystack experiment and Figure 4.

---

> > ### Author Response · Authors · 2023-01-27
> > **Response continued (2/4)**
> >
> > >The motivation behind this method is not completely clear. The paper provides two motivations: (1) Small amounts of crucial information might be ignored in the input; (2) we may want to improve an existing supervised model given extra inputs without retraining. Regarding (1), the synthetic datasets used in the experiments appear to exemplify the applications the authors have in mind, but it would be helpful to have a concise description of where this method should be applied. Reason (2) seems like an afterthought, and the TEB method isn't compared to simpler alternatives for this use case.
> >
> > For motivation (1): The estimation of transfer entropy has been utilized to analyze causality or connectivity of many real-world data streams (De La Pava Panche et al., 2019; Zhang et al., 2019; Lindner et al., 2019; Wu et al., 2021), and in principle, TEB would have many potential uses in domains similar to these, though for a different purpose (i.e. prediction as opposed to just measuring transfer entropy). In general, the use cases that are most relevant to TEB are those with small but important transfer entropy, which occurs when the target stream is well explained by an autoregressive model. In these cases, the potential contribution of the source stream to the prediction is limited relative to that of the autoregressive history, which is what makes capturing the transfer entropy challenging. Some examples of problems of this nature that TEB could be applied to include: 1. In finance, the prediction of one market index from one or more other indices. Depending on how closely these markets are connected economically, the directed information flow can be low relative to the history of the indices themselves, but still important to capture; 2. In neurophysiology, certain neural circuits are known to regulate physiological markers like heart rate (Rajendran et al. 2019), but these physiological signals are often periodic by themselves, so again, the important transfer entropy will be easily masked by the autoregressive correlations; 3. In weather forecasting, average monthly temperature in a region is likely stable with a yearly routine, but to detect anomalies deviating from such routine, additional sources of information might help such as air pressure, wind, greenhouse gas emission and other human activities; 4. In smart houses, the energy consumption of an electronic device at a given time period, say cell phone, can usually be tracked by its historical data, but the activities of other electronics could account for any out-of-ordinary cell phone usage, either longer or shorter.
> >
> > - (De La Pava Panche et al., 2019) De La Pava Panche, I et al. A Data-Driven Measure of Effective Connectivity Based on Renyi's α-Entropy. Frontiers in Neuroscience, 2019: 13
> > - (Zhang et al., 2019) Zhang, J J et al. Itene: Intrinsic transfer entropy neural estimator. ArXiv, abs/1912.07277, 2019.
> > - (Lindner et al., 2019) Lindner, B et al. Comparative analysis of granger causality and transfer entropy to present a decision flow for the application of oscillation diagnosis. Journal of Process Control, 2019: 79:72–84
> > - (Wu et al., 2021) Wu, Z et al. Effective Connectivity Extracted from Resting-State fMRI Images Using Transfer Entropy. Innovation and Research in BioMedical engineering (IRBM), 2021: 42(6): 457-465
> > - (Rajendran et al. 2019) Rajendran, P S et al. Identification of peripheral neural circuits that regulate heart rate using optogenetic and viral vector strategies. Nat Commun 10, 1944 (2019).
> >
> > We have added one of these practical examples to the text to clarify the first motivation in the “introduction” section, now the second paragraph becomes:
> >
> > “Estimating transfer entropy with a deep learning approach has been extensively explored ((Zhang et al. 2019) (see Appendix C for discussion contrasting with TEB). But, leveraging the transfer entropy for prediction is often a challenging problem and one that has yet to be resolved for deep neural networks (Schreiber, 2000; Gençağa, 2018). One reason for this difficulty is that the transfer entropy can be small relative to the overall entropy in the data streams, and as such, many prediction models will ignore it. **For example, in weather forecasting, periodicity of average monthly temperatures could dominate a prediction model of future temperatures, which would prevent a simple predictive model from detecting important anomalies indicated in other sources of information such as air pressure, wind, greenhouse gas emission and other human activities.** Indeed, we shall show that a neural network trained on a joint input stream can struggle to learn or generalize when the statistical coupling between the two processes is strongly correlated, but with a conditionally small and crucial information transfer from the source stream.”
> >
> > *response to this comment continued below...*

---

> > > ### Author Response · Authors · 2023-01-27
> > > **Response continued (3/4)**
> > >
> > > >The motivation behind this method is not completely clear. The paper provides two motivations: (1) Small amounts of crucial information might be ignored in the input; (2) we may want to improve an existing supervised model given extra inputs without retraining. Regarding (1), the synthetic datasets used in the experiments appear to exemplify the applications the authors have in mind, but it would be helpful to have a concise description of where this method should be applied. Reason (2) seems like an afterthought, and the TEB method isn't compared to simpler alternatives for this use case.
> > >
> > > Motivation (2) describes situations where one may want to incorporate new inputs while avoiding further training on an existing model. The reasons to include an existing fixed model are (but are not limited to): 1. It is a well-calibrated model with a good performance and/or interpretable output; 2. Retraining the model is computationally expensive and time-consuming; 3. The fixed model has already been deployed in practice and cannot be easily switched. Taking the above neurophysiology application as an example, neuroscientists may prefer to keep fixed a previously obtained heart rate autoregressive model because it is easy to analyze the effect of the neural signal on top of the autoregressive nature of the heart rate. For this motivation, we propose the TEB$^c$ formulation, and we demonstrated its performance in the rotating MNIST datasets. This proof-of-concept experiment demonstrates that TEB can work with a pre-trained model and that TEB is a valid candidate when facing the situation described in the second motivation. Moreover, in principle, we can repeat the process multiple times to iteratively incorporate more relevant variables into the model, so the TEB$^c$ formulation has lots of potential benefits in terms of flexibility.
> > >
> > > Moreover, as indicated in the rotating MNIST dataset, training TEB$^c$ is more difficult than TEB, as it imposes high requirements on the pre-trained Y module. To gain insights of it performing a more complicated task, we conducted additional experiments to compare TEB$^c$ to its deterministic version on the needle in the haystack task with 5 distractors, and we found the testing color accuracy of TEB$^c$ is 93.48 $\pm$ 3.93 %, while that of the deterministic baseline is 86.13 $\pm$ 4.67 %. Therefore, introducing an information bottleneck is even more advantageous with a fixed pre-trained Y module, compared to those without as reported in Table 1. One potential reason for the deterministic baseline’s poorer performance is that the pre-trained Y module might just not be good enough. An alternative reason could be that it is easier to overfit in this scenario, since here the deterministic model has all the information in the X stream just to find the color changes, without any form of regularization. We leave more comprehensive explorations, as well as comparison with other sophisticated baselines, for future work.
> > >
> > > We have modified the descriptions of the motivation of keeping a pre-trained model in the introduction section, now the last part of the third paragraph becomes:
> > >
> > > “As well, we design TEB in a manner that allows one to use a pre-trained predictive model for the target stream. That is, if one has an existing model that has been trained to predict $Y_t$ from the history of Y (i.e. $(Y_i)_{t-\ell\leq i<t}$), TEB allows one to plug this existing model into a new model that includes X in order to capture the transfer entropy and use it to improve the prediction of $Y_t$\. **This provides a flexible solution when one might have reasons to keep a previously trained model fixed. In practice, the reasons can be the desirable performance and interpretability of a given model, retraining being computationally expensive and time-consuming, and/or compatibility to other components in a larger system\.**”
> > >
> > > *response to this comment continued below...*

---

> > > > ### Author Response · Authors · 2023-01-27
> > > > **Response continued ((4/4)**
> > > >
> > > > >The motivation behind this method is not completely clear. The paper provides two motivations: (1) Small amounts of crucial information might be ignored in the input; (2) we may want to improve an existing supervised model given extra inputs without retraining. Regarding (1), the synthetic datasets used in the experiments appear to exemplify the applications the authors have in mind, but it would be helpful to have a concise description of where this method should be applied. Reason (2) seems like an afterthought, and the TEB method isn't compared to simpler alternatives for this use case.
> > > >
> > > > And we include the descriptions of the TEB$^c$ experiments on needle in the haystack task, in the Appendix B.3:
> > > >
> > > > “**Lastly, to demonstrate the performance of TEB$^c$ on this challenging dataset, we trained CEB with $\gamma=1$ on a separate dataset of 5 distractors with the bouncing balls colors never change for 1000 epochs, and used it as the pre-trained Y module to be incorporated into the TEB$^c$ formulation. This is in essence a simpler task than the normal needle in the haystack task, and the objective of the pre-trained Y module is to predict the localisation of the balls in the target frame while keeping the same color as in the histories. We trained both TEB$^c$ with $\beta=10$ and ‘’deterministic'' with the same pre-trained Y module fixed, on the 5 distractors dataset with 0.5 probability of switching the color, but both models suffered severe overfitting quickly. Therefore we increased the size of the dataset by $20$ times, and trained both models on it instead for 8 seeds. We found the testing color accuracy of TEB$^c$ is 93.48 $\pm$ 3.93 %, while that of the deterministic baseline is 86.13 $\pm$ 4.67 %. Therefore, introducing an information bottleneck is even more advantageous with a fixed pre-trained Y module, compared to those without as reported in Table 1. One potential reason for the deterministic baseline’s poorer performance is that the pre-trained Y module might just not be good enough for it. Alternative reason could be that it is still easier to overfit in this scenario, since here the deterministic model has all the information in the X stream just to find the color changes, without any form of regularizations. This demonstrates that TEB has the capability of incorporating a pre-trained model in this challenging task. However, training TEB$^c$ is considered as a more difficult task than TEB\.**”
> > > >
> > > > >It's not clear to me why the focus is on sequence data. Can't this method be applied to any supervised learning task where the inputs can be divided into a set of primary variables Y and secondary variables X, where X and Y are correlated but X might contain some key information? This seems to be a more general case that would be simpler to consider (especially since the experiments are on synthetic data anyway).
> > > >
> > > > Transfer entropy is a measure of directed information between two stochastic processes:
> > > >
> > > > $$T\_{X \overset{t}{\rightarrow} Y}= I(Y\_{t};(X\_{i})\_{t-\ell \leq i < t}|(Y\_{i})\_{t-\ell \leq i < t})$$
> > > >
> > > > Thus, transfer entropy is specifically concerned with sequence data. In non-sequential data the transfer entropy is the conditional mutual information.
> > > >
> > > > For example, if in the transfer entropy expression shown above, one just looks at a 1 time step sequence, it's essentially conditional mutual information between the 3 variables X,Y, and Y'. As such, and as noted in the paper, any derivation for transfer entropy will also  apply to this special case of conditional mutual information. As such, we agree with the reviewer that TEB could be empirically examined on 3 variable conditional mutual information data. However, since our focus is specifically on transfer entropy, which is ultimately concerned with sequence data, we consider that to be out of the scope of the current paper. Nonetheless, it could be an interesting future examination, and we thank the reviewer for pointing it out.
> > > >
> > > > >All experiments are on synthetic data sets that exhibit particular properties that the TEB aims to exploit.
> > > >
> > > > Our aim with this current work is to provide the analytical foundations of TEB and showcase its potential with synthetic data that allows easily interpretable results without specific domain expertise. As such, we prefer to leave more specialized applications that would require involved contextualisation, out of the scope of the current paper. However, we agree with the overall message from the reviewer and substantially improved our discussion and motivation surrounding real-world applications of this method. We leave exploration of TEB on real data sets as future work.

---

### Review · Reviewer_W7uf · 2023-01-06

**Summary Of Contributions:**

The paper proposes a 'Transfer Entropy Bottleneck' (TEB) method to train a latent variable model that can predict the future evolution of a target stream based on its own history and, at the same time, benefiting from information available in a coupled source stream. Importantly, the modelling is asymmetric - we only wish to predict the target stream, not the source. For this purpose the authors propose to use the transfer entropy (Schreiber, 2000) as a directional measure of information flow from source to target.

The TEB is formulated as a conditional latent variable model, building on previous work of Alemi et. al, 2017 (Variational Inforamtion Bottleneck, VIB) and mainly Fischer, 2020 (Conditional Entropy Bottleneck, CEB), the latter introducing the concept of Minimum Necessary Information (MNI) as an alternative to the classical Information Bottleneck (IB) principles (Tishby, 2020). The TEB extends MIN for the conditional setting (Conditional Minimum Necessary Information, CMNI) and in analog with CEB formulates the respective variational bound on TEB as the loss for the method.

The efficacy of the method is documented on a set of synthetic experiments, which illustrate the ability to transfer information from the source stream to inform the target stream prediction. The effect of the hyper-parameter choice in the loss on the transfer ability and the ability to reach the CMNI point is further discussed.
Real data experiments as well as discussion of theoretical guarantees of learnability are left for future work.

**Audience:**

Yes

**Broader Impact Concerns:**

I have no specific concerns here.

**Claims And Evidence:**

Yes

**Requested Changes:**

Major:
- Please add at a discussion about the similarities with and differences from the concept of Granger causality and mainly the large body of works that tries to learn this from the data.
- Theorems 1 and 2 are stated without any proofs. The only "proof" is the empirical evidence studied in the experiments. I do not think such an "anecdotal" evidence is sufficient for calling these statements theorems. Please either provide real mathematical proofs or rephrase avoiding the term "theorem".
- The discussion of Figure 4 is rather confusing. You mention $\beta_1$ and $\beta_0$ that as you say "we can identify" from the figure. I cannot and I do not see them anyhow identified in the figures directly.
- Figure 4 itself is confusing - green and red dashed lines should both indicate the true TE but the green is a horizontal line while red is vertical. I think, I am missing something - please correct or explain in description better.
- In section 3.2 under Figure 6 you explain the deterministic version of TEB. You say this has "no sampling". This is, however, the very first place in the paper where you mention some sampling. I can guess what sampling you mean (presumably z ~ q(z|x,y) using a standard reparametrization trick as in VAEs) but would be better not to guess. Please clarify.
- At the same place you say the model has "no KL". What does that mean? I am guessing this boils down to a simple AE architecture. Is this what you mean? Please clarify.

Other:
- Figure 4 and mainly Figure 9 is barely readable. Please improve! (Size, fonts, etc. Eg. Fig 9 could be split to 2 columns, etc.)


**Strengths And Weaknesses:**

Strengths - information theoric view of deep learning provides useful insights and an alternative justification for the latent variable formulation of deep learning (as a representation learning problem with compression motivated objectives). This paper builds on this theoretical framework to justify their model formulation, extending the existing work to the problem of stream / time/series data with asymmetric information transfer. This in my view is both interesting theoretically and potentially useful for practice. As far as I know, the model formulation is also new.

Weaknesses - a rather well-known concept related to asymmetric transfer of information in time-series forecasting is that of Granger causality. The concept and the possible relations to the proposed CMNI are not discussed in the paper at all. While I understand that the paper has to limit its scope somehow, I find that at least a brief discussion and comparison of the approaches is almost necessary in this context.

---

> ### Author Response · Authors · 2023-01-27
> **Response to Reviewer W7uf (1/4)**
>
> We thank the reviewer for the detailed review on our manuscript, please see our point-by-point response as below, please note that we highlight the text changes in the revision in bold:
>
> >Please add at a discussion about the similarities with and differences from the concept of Granger causality and mainly the large body of works that tries to learn this from the data.
>
> We agree with the reviewer that Granger causality is a term closely related with transfer entropy, and previous works have focused on their relationships. For example, Barnett et al. (2009) found that the two concepts are equivalent under a Gaussian assumption. Since reaching the CMNI point means reaching the true transfer entropy, there are indeed connections between Granger causality and our paper. To the best of our knowledge, the use of Granger causality lies generally in the causal analysis of the data, where the objective is to calculate the causal impact of one variable on another from data. We believe that this use of Granger causality puts it in a similar category as the estimation of transfer entropy.
>
> - (Barnett et al. (2009)) Barnett L et al. Granger causality and transfer entropy are equivalent for gaussian variables. Phys. Rev. Lett., 2009:103:238701
>
> However, we want to emphasize that the objective of TEB is not to estimate the transfer entropy from data, but rather to leverage the information embedded in the transfer entropy for better predictions. Although in principle, arriving at the CMNI point can give us the true transfer entropy, it is likely not possible, nor necessary, to precisely reach the CMNI point for a given dataset during the training of TEB. So our goal with TEB is not to provide another means of estimating transfer entropy, and hence, its goals are not comparable to the common use of Granger causality. In addition, estimating Granger causality of X to Y in practice  is usually based on linear vector autoregressive models, where two linear regression models are trained to predict Y with or without X as input. Although prediction models are also involved here, its limitation on the function form, i.e. the linear nature of the model, makes it also less relevant for comparison to TEB.
>
> Nonetheless, we agree with the reviewer that some discussion of Granger causality would be appropriate. We have added a paragraph at the end of section “related works” summarizing the above discussions with respect to the Granger causality:
>
> “**A related concept to transfer entropy is Granger causality (Granger, 1969). Different from transfer entropy, which can be applied to  non-linear models, Granger causality is formulated as linear autoregression, and many previous works have focused on the relationship between Granger causality and transfer entropy (Barnett et al., 2009, Hlavackova-Schindler, 2011). For example, Barnett et al. (2009) showed that the two concepts are equivalent under Gaussian assumptions. Calculating Granger causality is considered generally cheaper via linear vector autoregressive models (VAR) than transfer entropy. To estimate Granger causality of X to Y in practice, VAR trains two linear regression models to predict Y with or without X as input, and Granger causality can be calculated as a measure on how much the latter reduces the error of the former. However, to the best of our knowledge, the use of Granger causality is typically in the test of causality analysis, by calculating the measure from data of various domains, including neuroscience (Seth et al., 2015) and industrial processes (Lindner et al., 2019). This puts the related work utilizing Granger causality in a similar category as the estimation of transfer entropy. As noted, this diverges from the ultimate objective of TEB, which is to improve prediction in sequential data via transfer entropy. Lastly, although prediction models are also involved in VAR, its limitation on the function form, i.e. the linear nature of the regression model, makes it very different from TEB\.**”

---

> > ### Author Response · Authors · 2023-01-27
> > **Response continued (2/4)**
> >
> > >Theorems 1 and 2 are stated without any proofs. The only "proof" is the empirical evidence studied in the experiments. I do not think such an "anecdotal" evidence is sufficient for calling these statements theorems. Please either provide real mathematical proofs or rephrase avoiding the term "theorem".
> >
> > The proofs for both Theorem 1 and 2 are immediate from Proposition 3 and Lemma 5 that we have proved in the appendix. We have not made this clear in our original manuscript and we thank the reviewer for noting this omission. As follows, we describe the proofs for both theorems and add these clarifications to the revisions.
> >
> > For Theorem 1, with Proposition 3 proved in Appendix A.1, we have the following inequalities $I(Y';Z|Y)\leq I(Y';X|Y)$ and $I(Y';X|Y)\leq I(Z;X|Y)$. Thus we can arrive at the CMNI point by minimizing the following objective: $I(Z;X|Y) - \beta I(Y’;Z|Y)$, where $\beta>0$ is a common hyperparameter in IB objectives. In addition, we have Equation (3), (4) and (5) deriving the variational lower bound for $I(Y’;Z|Y)$ and upper bound for $I(Z;X|Y)$. Combining these, the TEB objective is immediate in the form indicated in Theorem 1.
> >
> > We have added the following sentence in the paper directly after Theorem 1:
> >
> > “**Proof. Directly from Proposition 3 in Appendix A.1, the CMNI point can be arrived at by maximizing $I(Y’;Z|Y)$ and minimizing $I(Z;X|Y)$. The proof of Theorem 1 is immediate from the proposition, by replacing $I(Y’;Z|Y)$ with its lower bound in Equation 4 and $I(Z;X|Y)$ with its upper bound in Equation 5. Please refer to Appendix A.1 and A.2 for the full derivations of Theorem 1\.**”
> >
> > For Theorem 2, directly from Lemma 5 in Appendix A.3, we have $I(C;Y’) = I(Y;Y’)$ gives  $I(X;Y’|C) = I(X;Y’|Y)$. So we can apply the same logic here as Theorem 1 with C replacing Y.
> >
> > We have added the following sentence in the paper directly after Theorem 2:
> >
> > “**Proof. Directly from Lemma 5 in Appendix A.3, with $I(C;Y’) = I(Y;Y’)$, we can replace Y everywhere with C and still arrive at the CMNI point. Then the proof is immediate by following a similar procedure as Theorem 1. Please refer to Appendix A.3 for the full derivations of Theorem 2\.**”
> >
> > For the complete proof regarding both TEB and TEB$^c$, please refer to Appendix A.
> >
> > >In section 3.2 under Figure 6 you explain the deterministic version of TEB. You say this has "no sampling". This is, however, the very first place in the paper where you mention some sampling. I can guess what sampling you mean (presumably z ~ q(z|x,y) using a standard reparametrization trick as in VAEs) but would be better not to guess. Please clarify.
> >
> > We thank the reviewer for catching the points of reduced clarity with respect to the deterministic baseline. The reviewer is correct that “no sampling” is indeed referring to the standard reparameterization trick applied to latent representation z. We have added the following sentence in to clarify:
> >
> > “**Then the reparameterization trick (Kingma & Welling, 2014) is used to sample latent representation Z from q(z|x,y)\.**”
> >
> > >At the same place you say the model has "no KL". What does that mean? I am guessing this boils down to a simple AE architecture. Is this what you mean? Please clarify.
> >
> > We apologize for the lack of clarity. By “no KL”, we refer to the fact that there is no KL divergence term in the loss function of the deterministic baseline. Structurally speaking, we agree with the reviewer that it is similar to an autoencoder, but the deterministic baseline has two encoders and one decoder, and its inputs and output are separate variables, which is different from the standard autoencoder used in unsupervised learning, whose inputs and outputs are the same variables.We have rephrased the description of the deterministic baseline as follows:
> >
> > “In our experiments, we also trained three baselines for comparison. The first two are deterministic baselines. Between them, the first is simply a deterministic version of the TEB model, with the same architecture and latent size\, **but we dropped the sampling from q(z|x,y) and excluded the KL divergence term in the objective, which is the first term in the TEB objective function. It uses directly the mean of q(z|x,y) as the input for the decoder\.**”
> >
> > >Figure 4 and mainly Figure 9 is barely readable. Please improve! (Size, fonts, etc. Eg. Fig 9 could be split to 2 columns, etc.)
> >
> > We have rearranged the layout of Figure 4 for better readability. We have also enlarged both the figure and legend size for Figure 9, as well as a yellow vertical line indicating the time of switch. Batch visualizations of prediction  in Figure 9 are now shown in 2 columns.

---

> > > ### Author Response · Authors · 2023-01-27
> > > **Response continued (3/4)**
> > >
> > > >The discussion of Figure 4 is rather confusing. You mention $\beta_1$ and $\beta_0$ that as you say "we can identify" from the figure. I cannot and I do not see them anyhow identified in the figures directly.
> > >
> > > We apologize for Figure 4 being confusing, and we thank the reviewer for pointing this out. Please note that both $\beta$s are empirically chosen from our observations. They are not well defined values, and are meant to just refer to certain phenomenons in our plots, and so, they are subject to change in different datasets. Between the two, $\beta_1$ is referred to as the turning point where for $\beta<\beta_1$ the output loglikelihood sharply drops off, and we can approximately identify it from the loglikelihood subfigure in Figure 4. $\beta_0$ is referred to as the situation where the model is not learning from the X stream, and it approximately corresponds to zero $I(Z;X|Y)$. Therefore for TEB, we can find in Figure 4a that $\beta_1$ is around 0.2 and $\beta_0$ is between 0.1 and 0.14. For TEB$^c$, we can find in Figure 4b that $\beta_1$ is approximately in the neighborhood of 0.2, but for $\beta_0$, we need to look also at Figure 5b to find that it is around 0.1.
> > >
> > > We have rephrased the paragraph describing these points as follows:
> > >
> > > “**From Figure 4a, we can see that there is a "bifurcation point’’ for $\beta$ around 0.2, such that for any $\beta$ smaller than that the output loglikelihood sharply drops off. We refer to such a point as $\beta_1$, and the sharp drop-off for $\beta<\beta_1$  is likely due to that essential information transfer from X being left out of the latent representation Z of the model, as in $I(Z;X|Y)<I(Y';X|Y)$. On the other hand, as in other IB methods (Wu et al., 2019), there is another value of $\beta$, which we denote $\beta_0$ as in Wu et al. (2019), such that for $\beta<\beta_0$ the model is not learning and Z becomes a trivial representation, which in our case means that we compressed too much on the information in the bottleneck, so that the information from X to recover Y' is not available to the model any more. From Figure 4a, we can put $\beta_0$ roughly in the range from 0.1 to 0.14, corresponding to nearly zero $I(Z;X|Y)$\.**”
> > >
> > > And for TEB$^c$:
> > >
> > > “**We find in Figure 4b that the bifurcation point $\beta_1$ for TEB$^c$ is similar to that recovered for TEB trained end to end. However, we do find that unlike the situation for TEB, here the CMNI point does not appear to be reachable since our bound for $I(Y';Z|Y)$ continues to increase for $\beta>\beta_1$. On the other hand, It is difficult to find $\beta_0$ directly from Figure 4b, but as shown in Figure 5b, TEB$^c$ fails to predict the change of digit class at $\beta=0.1$, which puts its $\beta_0$ approximately at 0.1\.**“

---

> > > > ### Author Response · Authors · 2023-01-27
> > > > **Response continued (4/4)**
> > > >
> > > > >Figure 4 itself is confusing - green and red dashed lines should both indicate the true TE but the green is a horizontal line while red is vertical. I think, I am missing something - please correct or explain in description better.
> > > >
> > > > Although they both relate to the true transfer entropy, the green horizontal line corresponds to the true transfer entropy, while the red vertical line corresponds to the specific beta that achieves the true transfer entropy. Note that for both TEB and TEB$^c$ testing performance in Figure 4, the left subfigure is $I(Z;X|Y)$ vs $\beta$, in which the green vertical line indicates the specific value of $\beta$ ($\beta^{\*}$) that achieves the true transfer entropy, and we find $\beta^{\*}$ by interpolating the average obtained $I(Z;X|Y)$ from different values of $\beta$. The right subfigure is output loglikelihood vs $\beta$, where the red vertical line is drawn to show the reconstruction performance associated with $\beta^{\*}$, hence the *true TE @ $\beta^{\*}$*. Therefore, the two dashed lines in combination are used to demonstrate the reconstruction performance when the model achieves the true transfer entropy. For TEB, $\beta^{\*}$ also has approximately the best testing reconstruction performance, but for TEB$^c$, although $\beta^{\*}$ is not among the best-performing ones, it is still a bifurcation point in the output loglikelihood curve. Lastly, recall that we are interested in the $\beta$ achieving the maximal testing reconstruction performance as it gives us $I(Y’;Z|Y) \approx I(Y’;X|Y)$, since output loglikelihood is an lower bound of $I(Y’;Z|Y)$.
> > > >
> > > > We have modified the caption of Figure 4 to reflect the explanation provided above. We hope the updated text clarifies the plot.
> > > >
> > > > “**Plots of the information metric and reconstruction loglikelihood vs $\beta$ incremented by tenths, with extra hundredths increments in the chaotic regime in $[.1,.2]$, for both TEB and TEB$^c$ testing. In the information metric subfigure (the left subfigure in both (a) and (b)), the green dashed horizontal line indicates the true transfer entropy for this dataset. From this we obtain $\beta^{\*}$ as the specific value of $\beta$ achieving the true transfer entropy, calculated as the interpolated value of $\beta$ for which $I(Z;X|Y)=I(Y';X|Y)$ on average from repeated trials. In the reconstruction loglikelihood subfigure (the right subfigure), the output loglikelihood of the model trained with the obtained $\beta^{\*}$ is marked using the red dashed vertical line. Note that in (a) the variance of runs highly increases for $\beta\in(.1,.2)$, but the models in this interval that found a solution with $I(Z;X|Y)\approx I(Y';X|Y)$ are the same models that achieved higher reconstruction performance\.**”

---

> > > > > ### Comment · Reviewer_W7uf · 2023-01-28
> > > > > **Proofs, better comments for plots and some rephrasing - improved paper**
> > > > >
> > > > > Dear authors,
> > > > > thank you for introducing all the clarification updates. I think it improved the paper.

---

> > ### Comment · Reviewer_W7uf · 2023-01-28
> > **Nonlinear Granger causality for forecasting**
> >
> > Dear authors, thank you for your effort to complement your paper with these links. However, I do not quite agree with your statements neither about the "purpose" of Granger modelling nor the "linearity" of the models. See for example https://link.springer.com/chapter/10.1007/978-3-319-71246-8_33. While I understand that a full analysis of the links may be asking for a completely new paper (along the lines of Barnett et al 2009), I believe at least a brief recount of the relationships would be beneficial.

---

> > > ### Author Response · Authors · 2023-01-30
> > > **Update on the discussion of Granger causality models.**
> > >
> > > We thank the reviewer for sharing with us a relevant paper leveraging Granger causality in forecasting in a non-linear way. We have updated the corresponding paragraph in section “‘related works” to discuss those methods and highlight the difference of these methods from IB formulations. We have re-uploaded the revision, and now the paragraph reads as
> > >
> > > “**Another related concept to transfer entropy is Granger causality (Granger, 1969), and many previous works have focused on the relationship between the two (Barnett et al., 2009; Hlavackova-Schindler, 2011). For example, Barnett et al. (2009) showed that the two concepts are equivalent under Gaussian assumptions. Originally developed in econometrics, Granger causality has gained attention in other domains, including neuroscience (Seth et al., 2015) and industrial processes (Lindner et al., 2019). The calculation of Granger causality is traditionally via linear vector autoregressive models (VAR), which is generally considered cheaper than transfer entropy. To estimate Granger causality of X to Y in practice, VAR trains two linear regression models to predict Y with or without X as input, and Granger causality can be calculated as a measure on how much the latter reduces the error of the former. Throughout the years, many works extended this traditional approach in terms of both causality analysis from the data and time-series forecasting, for the linear model (Lozano et al., 2009), as well as non-linear models including kernel regression (Gregorová et al., 2017), radial basis functions neural network (Wismüller et al., 2021), multilayer perceptrons (Talebi et al., 2018; Tank et al., 2022) and recurrent neural networks (Tank et al., 2022). In general, these methods aim to learn the causal relationships among different time-series and / or among different time-lagged variables in a regression problem, and that is very different from the objective of TEB, and other IB methods described above, which is to learn a compressed latent representation of the data\.**”

---

> > > > ### Comment · Reviewer_W7uf · 2023-01-30
> > > > **Granger - yes, gives better picture**
> > > >
> > > > Dear authors, thanks you for this para, I find it provides a useful broader picture of the related concepts.
> > > > I would still like to see how these two concepts really link to each other but that's indeed outside the scope of your paper.
> > > > Perhaps it is a research direction I shall undertake :)
> > > > Thanks.

---

> > > > > ### Author Response · Authors · 2023-01-30
> > > > > **Thank you for the reply**
> > > > >
> > > > > Dear reviewer, we are glad to hear that our updated text gives better picture of the concepts. Indeed, this is an interesting research direction. Again, we thank the reviewer for engaging with us in this discussion and for the useful comments that improve our paper.

---

### Review · Reviewer_k9TR · 2023-01-17

**Summary Of Contributions:**

The paper derives an information bottleneck objective between separate data streams using the concept of transfer entropy (Transfer Entropy Bottleneck).

It validates its approach empirically in several settings and analyzes the behavior of its objective for different strengths of the bottlenecks ($\beta$ hyperparameter).

**Audience:**

Yes

**Broader Impact Concerns:**

The broader impact statement might be removed (I do not think it is necessary for TMLR, and its current form is laudable but very unspecific---I'm not faulting that).

**Claims And Evidence:**

Yes

**Requested Changes:**

In addition to the clarifications and suggestions in the previous section, the paper might be improved by slight re-layouting. Some of the figures might be rotated to save space (5, 7, , while other figures could be enlarged (e.g. 1ab, 3ab, 8, 9).

Page 7 and 8 also seem to have a lot of unnecessary whitespace from formatting issues.

**Strengths And Weaknesses:**

The paper is great as-is. It is very well written and clear in its exposition. Its literature review introduces CEB and VIB nicely.

The experiments are clear and show that the TEB works as intended in the presented settings. I especially like the introductory experiment §3.1 and Fig 4.a) which clearly shows the $\beta$ trade-off and takes time to explain the considerations.

As such, I think the paper will be both interesting to readers, and its claims are clear and accurate as provided.

The appendix is also extensive and contains details for reproduction as well as detailed proofs.

### Regarding the TEB:
Please correct me if I'm wrong, but as far as I understand, the data processing inequality also applies to the conditional quantities, as such, some of the proofs might not be necessary:

In particular, for any $y$, we can look at the probability space $p(y',x,z \mid y)=p(y' \mid x, y) \, p(z \mid x, y)$. Within this probability space, the DPI holds without change; that is, we have:
$$I(Y'; Z \mid y) \le I(Y' ; X \mid y) \land I(Y';X \mid y) \le I(Z ; X \mid y).$$
Taking an expectation over $y$ (and continuity) provides the result.

Can the TEB can also be found from a decomposition of a joint IB between $(Y,X)$ and $Y'$:
$$
\underbrace{I(Y,X;Y')}\_{\ge I(C, Z; Y')} =
\underbrace{I(Y; Y')}\_{\ge I(C; Y')} +
\underbrace{I(X; Y' \mid Y)}\_{I(\ge Z; Y' \mid Y)}?
$$

### Haystack

From my reading, the haystack experiment only uses $X$ as input for the deterministic joint and CEB baselines. Even though $X$ is considered to be unified in the sense that it contains all the information that $Y$ contains, isn't this a harder task? When both $X$ and $Y$ are provided, to find the needle, we only need to subtract $X-Y$. Could this explain the bad performance of the joint baselines? If this is obvious, it was not made clear, and if I misunderstood, the text might benefit from further clarification.

In that regard, could an example be provided of X and Y in figure 6?

---
Lastly (but independently of the review), will example notebooks or similar be released/

---

> ### Author Response · Authors · 2023-01-27
> **Response to Reviewer k9TR (1/2)**
>
> We thank the reviewer for the detailed review on our manuscript, please see our point-by-point response as below, please note that we highlight the text changes in the revision in bold:
>
> >In addition to the clarifications and suggestions in the previous section, the paper might be improved by slight re-layouting. Some of the figures might be rotated to save space (5, 7, , while other figures could be enlarged (e.g. 1ab, 3ab, 8, 9).
>
> >Page 7 and 8 also seem to have a lot of unnecessary whitespace from formatting issues.
>
> We thank the reviewer for the suggestions. We have made modifications targeting these layout and formatting issues. Note that we didn’t rotate Figure 5, because we want to keep the same column layout for consistency throughout the paper. Please refer to our revision for more details.
>
> >Regarding the TEB: Please correct me if I'm wrong, but as far as I understand, the data processing inequality also applies to the conditional quantities, as such, some of the proofs might not be necessary:
>
> >In particular, for any y, we can look at the probability space p(y’,x,z|y)=p(y’|x,y)p(z|x,y). Within this probability space, the DPI holds without change; that is, we have: I(Y’;Z|y) <= I(Y’;X|y) and I(Y’;X|y) <= I(Z;X|y). Taking an expectation over y (and continuity) provides the result.
>
> We thank the reviewer for pointing out DPI also holds for the conditional quantity under specific probabilistic graphs.  However for TEB, the probabilistic graphs we assume are different, as shown in Equation (1) and (2), so we provide a simple proof on these inequalities in our case, summarized in Proposition 3. We include them with the intention to keep the completeness in our derivations.
>
> >Can the TEB can also be found from a decomposition of a joint IB between (Y,X) and Y':
>
> >$$\underbrace{I(Y,X;Y')}\_{\geq I(C,Z;Y')} = \underbrace{I(Y;Y')}\_{\geq I(C;Y')}+ \underbrace{I(X;Y'|Y)}\_{\geq I(Z;Y'|Y)}$$
>
> We apologize but we are not certain we understand what the reviewer is looking for here. Could the reviewer expand on what is implied by the statement? For example, what is the specific probabilistic graph under the chain rule? And what is C referring to, as TEB is mentioned here instead of TEB$^c$. In particular, it seems that an IB between (Y,X) and Y' would refer to the markov chain (Y,X) ->Z -> Y', which is different from TEB. There are many links between several information theoretic quantities, several of which are outlined in the cited references detailing TEB. Should the reviewer believe that a specific link would considerably improve the message surrounding our proposed ML method, we would be happy to include it in the discussion.

---

> > ### Author Response · Authors · 2023-01-27
> > **Response continued (2/2)**
> >
> > >Haystack
> >
> > >From my reading, the haystack experiment only uses X as input for the deterministic joint and CEB baselines. Even though X is considered to be unified in the sense that it contains all the information that Y contains, isn't this a harder task? When both X and Y  are provided, to find the needle, we only need to subtract X-Y. Could this explain the bad performance of the joint baselines? If this is obvious, it was not made clear, and if I misunderstood, the text might benefit from further clarification.
> >
> > We thank the reviewer for catching this point. Indeed, it is harder to learn from the joint input stream here, because it cannot utilize a certain relationship between X and Y streams, which in turn can be as helpful as the semantic subtraction in this task. That could be one explanation of their bad performance. Unfortunately, we cannot directly concatenate X and Y as the joint input in this task, because that would give every distractor a duplication and make identifying the needle pixel as trivial as filtering out all the pairs. Despite that, we still believe it is worth including the joint stream baselines, as they contain all the information with respect to the distractors and needle pixel. Moreover, even with a split-path of X and Y streams, as the model is not allowed to do the subtraction in the original image spaces, learning to do the "subtraction’’ is still not trivial. In fact, as combining information from X and Y occurs only at Z, the "subtraction’’ would have to be done in the latent space. All in all, we believe this is still a challenging task for TEB, as it needs to infer the needle pixel from just X, based on the ball color in the past 3 frames. We show that TEB is advantageous in inferring the needle pixel by comparing it to the deterministic baseline, with an increasing number of distractors.
> >
> > We believe that another explanation lies in the joint stream formulation. Take CEB as examples, the data being compressed in its bottleneck is the joint stream data as a whole, whereas TEB compresses the information from the X stream based on what information is contained in the Y stream. This is the directed way of doing the compression, making it easier to capture the transfer entropy which is also directed.
> >
> > We have modified discussions in Section 3.2, now it reads as follows:
> >
> > “**In comparison, both the joint stream deterministic and stochastic models (CEB) do not even learn to generalize better than the constant color baseline on 5 distractors. One explanation is that they are likely dealing with a harder task than TEB and "Deterministic'', because they cannot make use of the difference from X to Y in the latent space, which is essentially the information denoting the needle pixel. Another explanation lies in the joint stream formulation. For example for CEB, it has to compress the joint stream input as a whole, including the bouncing balls images, needle pixel and all the distractors, so it might ignore the needle pixel which is relatively small\.**”
> >
> > >In that regard, could an example be provided of X and Y in figure 6?
> >
> > Strictly within the scope of Figure 6, that is, with the only first 10 distractors shown and randomly shifted on the image perimeters, X would be the images as they are from time step t-3 to t-1, and Y would be X but without the needle pixels at the top-left corners of the images. Please note that Figure 6 is only for visualization and is not exactly what we used in our experiments, and the actual X and Y streams are stackings of different RGB channel groups. Please see Figure 17 in Appendix B.3 for more information.
> >
> > >Lastly (but independently of the review), will example notebooks or similar be released
> >
> > Right now we only have the code in the git repository, but in the future, we do have plans to provide example notebooks demonstrating the datasets and sample generations with a trained model.

---

### Author Response · Authors · 2023-01-27
**Revision**

We thank all the reviewers for their constructive reviews. We have updated the paper based on your comments and suggestions, and the changes have been marked in red. Below we have provided individual responses to specific points raised in the review, along with the corresponding modification we make in the paper revision, but we would like to summarize the key parts here for better clarifications:

1. We have fixed some layout issues of the figures. Some of the figures are too small, and we have enlarged them for better readability.
2. We have added to the introduction an application of the TEB method, and more descriptions on the motivation of using a pre-trained model.
3. We have added a paragraph in related work on Granger causality.
4. We have modified the unclear wordings with respect to the conditional independence. We have also added descriptions on the proofs of Theorem 1 & 2.
5. We have updated the caption of Figure 4, along with the related discussion on it, for better clarifications.
6. We have updated Section 3.2, with better descriptions on the baselines and a discussion on potential reasons for joint stream models’ poor performance.
7. We have conducted new experiments with pre-trained model on needle in the haystack task, and added the descriptions and results in Appendix B.3.

We hope that the revision and responses have addressed your concerns.

---

### Author Response · Authors · 2023-02-28
**Camera ready version uploaded**

We have uploaded the camera-ready version of our paper. We would like to thank the anonymous reviewers and the action editor for their valuable feedback and discussion, which definitively leads to improvements of the paper!

---

### Decision · Action_Editors · 2023-02-17

**Recommendation:** Accept as is

**Comment:**

The reviewers have unanimously consented that the paper validates its claims and is of interest to the community.

In particular the reviewers feel as though their concerns have been addressed in the updated version.

I want to thank the reviewers and authors for taking the time and engaging in the process, seems the paper has benefited as a result.

**Audience:**

The paper has a clear audience.

**Claims And Evidence:**

With the reviewers in unanimous agreement, I believe the paper supports its claims with accurate and convincing evidence.